# Aβ(1-42) tetramer and octamer structures reveal edge conductivity pores as a mechanism for membrane damage

Sonia Ciudad [1,2,11], Eduard Puig[1,2,3,11], Thomas Botzanowski[4], Moeen Meigooni[5], Andres S. Arango[5], Jimmy Do[5], Maxim Mayzel [6], Mariam Bayoumi[7], Stéphane Chaignepain[1], Giovanni Maglia [8], Sarah Cianferani[4], Vladislav Orekhov[6,9], Emad Tajkhorshid [5], Benjamin Bardiaux [10] & Natàlia Carulla [1,2 ✉]

Formation of amyloid-beta (Aβ) oligomer pores in the membrane of neurons has been proposed to explain neurotoxicity in Alzheimer's disease (AD). Here, we present the three-dimensional structure of an Aβ oligomer formed in a membrane mimicking environment, namely an Aβ(1-42) tetramer, which comprises a six stranded β-sheet core. The two faces of the β-sheet core are hydrophobic and surrounded by the membrane-mimicking environment while the edges are hydrophilic and solvent-exposed. By increasing the concentration of Aβ(1-42) in the sample, Aβ(1-42) octamers are also formed, made by two Aβ(1-42) tetramers facing each other forming a β-sandwich structure. Notably, Aβ(1-42) tetramers and octamers inserted into lipid bilayers as well-defined pores. To establish oligomer structure-membrane activity relationships, molecular dynamics simulations were carried out. These studies revealed a mechanism of membrane disruption in which water permeation occurred through lipid-stabilized pores mediated by the hydrophilic residues located on the core β-sheets edges of the oligomers.

[1] University of Bordeaux, CBMN (UMR 5248)—CNRS—IPB, Institut Européen de Chimie et Biologie, 2 rue Escarpit, 33600 Pessac, France. [2] Institute for Research in Biomedicine (IRB Barcelona), The Barcelona Institute of Science and Technology (BIST), Baldiri Reixac 10, 08028 Barcelona, Spain. [3] Departament de Química Inorgànica i Orgànica, Universitat de Barcelona, Martí i Franqués 1, 08028 Barcelona, Spain. [4] Laboratoire de Spectrométrie de Masse BioOrganique, Université de Strasbourg, CNRS UMR7178, IPHC, Strasbourg, France. [5] NIH Center for Macromolecular Modeling and Bioinformatics, Beckman Institute for Advanced Science and Technology, Center for Biophysics and Quantitative Biology and Department of Biochemistry, University of Illinois at Urbana-Champaign, Urbana, IL 61801, USA. [6] Swedish NMR Centre, University of Gothenburg, Box 465, 405 30 Gothenburg, Sweden. [7] Biochemistry, Molecular and Structural Biology Section, University of Leuven, Celestijnenlaan 200G, 3001 Leuven, Belgium. [8] Groningen Biomolecular Sciences & Biotechnology Institute, University of Groningen, 9747 AG Groningen, The Netherlands. [9] Department of Chemistry and Molecular Biology, University of Gothenburg, Box 465, 405 30 Gothenburg, Sweden. [10] Structural Bioinformatics Unit, Department of Structural Biology and Chemistry, C3BI, Institut Pasteur; CNRS UMR3528; CNRS USR3756, Paris, France. [11] These authors contributed equally: Sonia Ciudad, Eduard Puig. ✉email: natalia. carulla@gmail.com

Substantial genetic evidence links the amyloid-β peptide (Aβ) to Alzheimer's disease (AD)[1]. However, there is great controversy in establishing the exact Aβ form responsible for neurotoxicity. Aβ is obtained from a membrane protein, the amyloid precursor protein (APP), through the sequential cleavage of β- and γ-secretase[2]. Upon APP cleavage, it is generally considered that Aβ is released to the extracellular environment. Due to its hydrophobic nature, Aβ then aggregates into multiple species, commonly referred to as soluble Aβ oligomers, which eventually evolve into Aβ fibrils[3–6], the main component of amyloid plaques. Moreover, there is a less described pathway that considers that upon APP cleavage, a fraction of Aβ remains in the membrane evolving into membrane-associated Aβ oligomers, which would be directly responsible for compromising neuronal membrane integrity[7].

Since the amounts of Aβ fibrillar plaques do not correlate with cognitive decline[8] researchers have focused on the study of both soluble and membrane-associated Aβ oligomers to identify the Aβ form responsible for neurotoxicity. Soluble Aβ oligomers have been prepared incubating synthetic Aβ samples under specific conditions hypothesized to stabilize a given Aβ oligomer form (culture media, pH, T, salts, …) or engineering Aβ variants to lock the peptide in a conformation that is incompatible with fibril formation[9]. Many types of Aβ oligomers, such as ADDLs, amylospheroids, paranuclei or hexameric Aβ42cc[9] have been prepared using these approaches. In a similar manner, research dedicated to study Aβ in a membrane environment has used either detergent micelles[10–12] or liposomes[13–16]. The study of Aβ in the presence of liposomes has been imaged by atomic force microscopy (AFM) revealing that Aβ incorporates into liposomes as oligomeric pores of different sizes[15]. Moreover, functional characterization of these samples using electrophysiological recordings in lipid bilayers demonstrated the presence of multiple single-channel currents of various sizes[13–15]. These results led to the proposal of the amyloid pore hypothesis nearly three decades ago[13].

However, in spite of the many efforts in this area, none of these studies have provided atomic structures for any Aβ oligomer. Without this information, it has not been possible to unequivocally establish Aβ oligomers' mechanism of neurotoxicity or to design therapeutic strategies against their neurotoxic effects[17]. In 2016, we reported conditions to prepare homogenous and stable Aβ oligomers in membrane-mimicking environments[18]. We found that their formation was specific for Aβ(1-42)—the Aβ variant most strongly linked to AD—, that they adopted a specific β-sheet structure, which is preserved in a lipid environment provided by bicelles, and that they incorporated into membranes exhibiting various types of pore-like behavior. Because of these properties, we named them β-sheet pore-forming Aβ(1-42) oligomers (βPFOs$_{Aβ(1-42)}$). Here we present the atomic structures of βPFOs$_{Aβ(1-42)}$ by nuclear magnetic resonance (NMR) and mass-spectrometry (MS) and provide a mechanism for membrane disruption based on electrophysiology and simulation studies in membranes.

## Results

**βPFOs$_{Aβ(1-42)}$ sample comprise Aβ(1-42) tetramers.** In an earlier study, when developing conditions to prepare βPFOs$_{Aβ(1-42)}$, we aimed at characterizing biologically relevant Aβ oligomers so we established conditions for their formation while working at pH 7.4[18,19]. However, we also found that the oligomers adopted the same structure while being more stable when prepared at pH 9.0 (Supplementary Fig. 1). Since structural characterization of βPFOs$_{Aβ(1-42)}$ was facilitated when working with stable samples, we decided to work at pH 9.0. We prepared a selectively labeled

$^2$H,$^{15}$N,$^{13}$C βPFO$_{Aβ(1-42)}$ sample in dodecylphosphocholine (DPC) micelles at pH 9.0 and used high field NMR triple-resonance TROSY-type experiments to obtain sequence-specific resonance assignments (Supplementary Figs. 2 and 3). Peak assignment allowed us to establish that Aβ(1-42) residues were observed in duplicate in the 2D [$^1$H,$^{15}$N]-TROSY spectrum (Fig. 1a), which suggested that the sample comprised two distinct Aβ(1-42) subunits. To highlight the detection of two Aβ(1-42) subunits in the sample, residues belonging to each of them were identified as either red or green. Next, we used the Cα and Cβ chemical shifts to derive the three-residue averaged (ΔCα-ΔCβ) secondary chemical shifts to thus determine the presence of secondary structure elements in each Aβ(1-42) subunit (Fig. 1b). This analysis revealed that the red Aβ(1-42) subunit contributed two β-strands, β1 and β2, to the oligomer structure. These strands extended, respectively, from G9 to A21 and from G29 to V40. Instead, in the green Aβ(1-42) subunit, residues L17 to F20 exhibited α-helical propensity, while residues G29 to I41 adopted a β-strand conformation, referred to as α1 and β3, respectively. To finalize assignments, the connectivity between β1 and β2, and α1 and β3 secondary structural elements was established using mixtures of Aβ(1-42) and Aβ(17-42) with distinct isotope labels (Supplementary Fig. 4).

Next, we used nuclear Overhauser effect spectroscopy (NOESY) to obtain long-range structural information. From the cross-peaks observed in the 3D NH-NH NOESY experiment, we identified eight NOEs between β1 and β2 strands of the red Aβ(1-42) subunit and 7 NOEs between β2 strand of the red Aβ(1-42) subunit and the β3 strand of the green Aβ(1-42) subunit (Fig. 1c). The observation of intra- and inter-subunit NOEs allowed us to establish the topology of an asymmetric dimer unit and to confirm that all the peaks detected in the 2D [$^1$H,$^{15}$N]-TROSY spectrum belonged to the same oligomer. Moreover, we also detected three NOEs involving residues of the β3 strand (Fig. 1c), which could be explained only if two asymmetric dimer units interacted through β3 to form a tetramer in an antiparallel manner. All together, these NOEs allowed us to establish the complete topology of a six-stranded Aβ(1-42) tetramer unit (Fig. 1d). Moreover, since we did not detect any NOEs for the amide protons of β1 residues pointing outward of the β-sheet core (i.e., Y10, V12, H14, K16, V18, and F20), we inferred that the signals detected by NMR corresponded to an Aβ(1-42) tetramer. To further validate the tetramer topology, we prepared specifically isotope-labeled samples and assigned the methyl groups of Ala, Ile, Leu, and Val (AILV) residues (Supplementary Fig. 5). We then acquired 3D NH-CH$_3$ NOESY and 3D CH$_3$-CH$_3$ NOESY spectra and obtained a network of 87 NH-CH$_3$ and 25 CH$_3$-CH$_3$ NOEs consistent with the topology of the tetrameric unit (Supplementary Fig. 6).

NMR NOESY-type experiments allowed us to identify a network of more than 150 NOE contacts (Supplementary Table 1) which, together with backbone dihedral angle (Supplementary Fig. 7) and hydrogen-bond restraints, allowed us to define the structure of an Aβ(1-42) tetramer (Fig. 2a, Supplementary Fig. 8a, b and Supplementary Table 2). The tetramer comprised a β-sheet core made of six β-strands, connected by only two β-turns, leaving two short and two long, flexible N-termini, the latter comprising α1. The root mean square deviation (RMSD) of the Aβ(1-42) tetramer ensemble was 0.77 and 1.34 Å for the backbone and the heavy atoms of the six-stranded β-sheet core, respectively. Notably, all residues on both faces of the β-sheet core were hydrophobic except for three basic residues (i.e., H13, H14, and K16) located in β1, at the edges of the β-sheet core (Fig. 2b). On the other hand, residues making the β-turns and the ends of the flexible N-termini were hydrophilic except for those comprising α1.

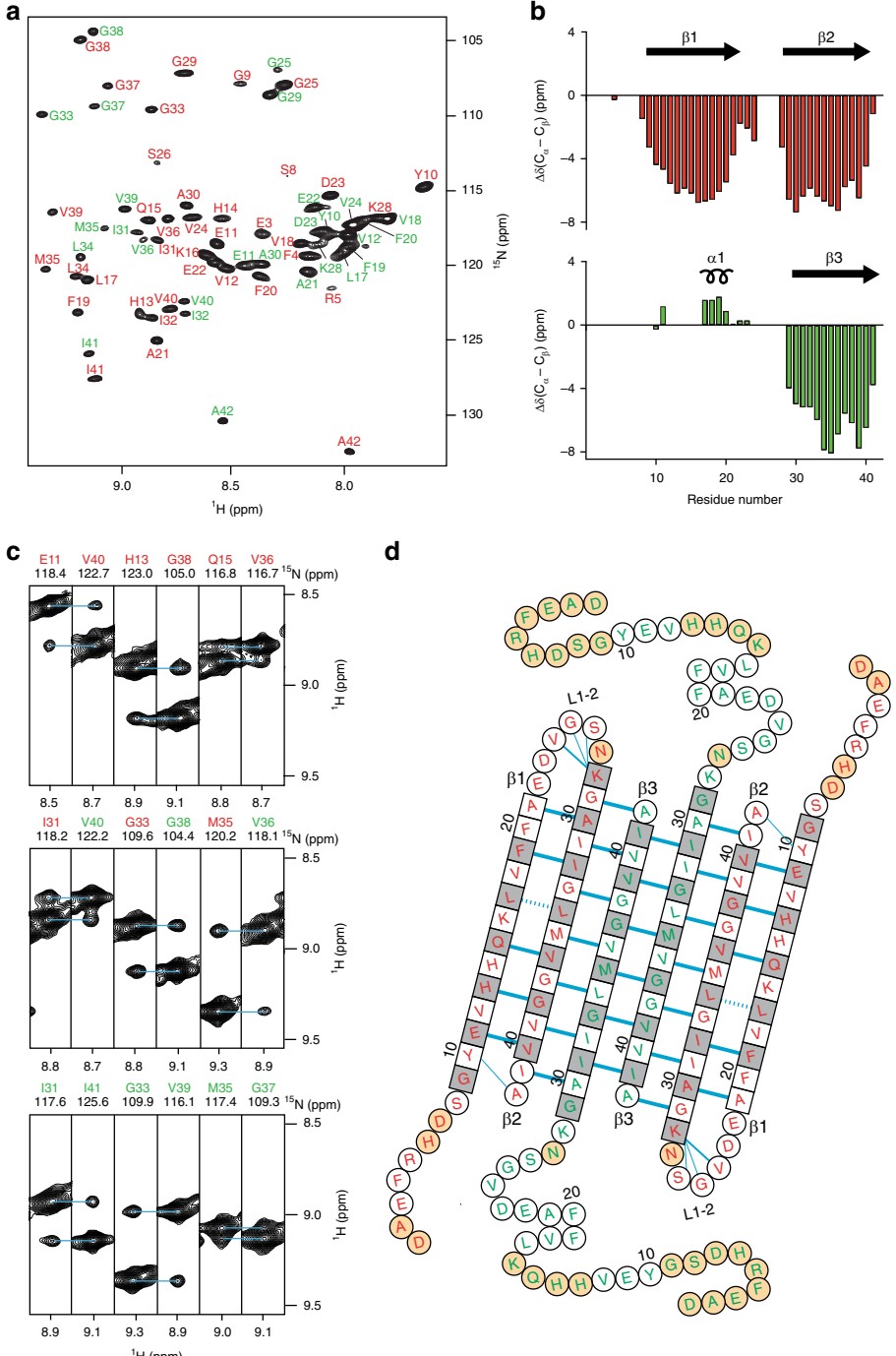

**Fig. 1 Architecture of the Aβ(1-42) tetramer. a** Amide resonance assignments of the Aβ(1-42) tetramer. Two Aβ(1-42) subunits are detected and residues belonging to each of them are labeled in either red or green. **b** Three-bond averaged secondary chemical shifts versus residue number for the red (top) and the green (bottom) Aβ(1-42) subunits. Secondary structural elements derived from chemical shift indices are shown at the top with its corresponding number. Arrows indicate β-strands and helical symbols helices. **c** Strips from a 3D NH-NH NOESY spectrum defining long-range intra-monomer interactions between the red Aβ(1-42) subunit, long-range inter-monomer interactions between the red and the green Aβ(1-42) subunits, and long-range inter-dimer interactions between the two green Aβ(1-42) subunits. **d** The amino acid sequence of the Aβ(1-42) tetramer is arranged on the basis of the secondary and tertiary structure. Amino acids in square denote β-sheet secondary structure as identified by secondary chemical shifts; all other amino acids are in circles. Blue lines denote experimentally observed NOE contacts between two amide protons. Bold lines indicate strong NOEs typically observed between hydrogen-bonded residues in β-sheets. Dashed lines show probable contacts between protons with degenerate ¹H chemical shifts. The side chains of white and gray residues point towards distinct sides of the β-sheet plane, respectively. Orange circles correspond to residues that could not be assigned. Sample conditions were 1 mM ²H,¹⁵N,¹³C Aβ(1-42) in 10 mM Tris, 28.5 mM DPC at pH 9.0 after incubation for 24 h at 37 °C. Source data are provided as a Source data file.

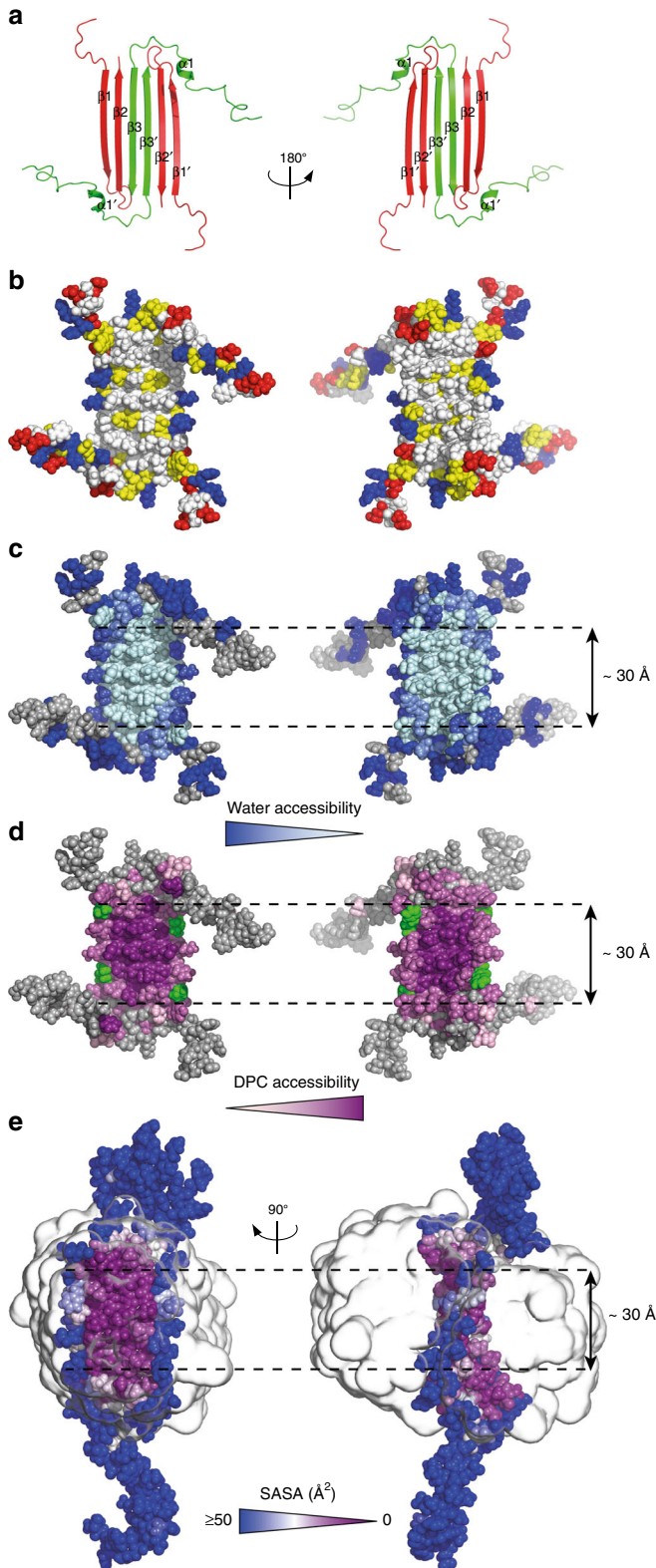

**Fig. 2 3D structure of the Aβ(1-42) tetramer prepared in DPC. a** Ribbon diagram of the Aβ(1-42) tetramer structure. Aβ(1-42) subunits are colored either red or green to identify the asymmetric dimer unit that constitutes the building block of the Aβ(1-42) tetramer. **b** Distribution of hydrophobic and charged residues on the surface of the Aβ(1-42) tetramer. Hydrophobic residues are white, polar are yellow, and positively and negatively charged are red and blue, respectively. **c** Water accessibility of amide protons revealed through 2D [$^1$H,$^{15}$N]-HSQC spectra obtained at different pHs and through measurement of amide temperature coefficients. Solvent accessibility is linearly coded on the basis of the intensity of blue, with light blue corresponding to low water accessibility and dark blue corresponding to high water accessibility. Unassigned residues are shown in gray. **d** DPC accessibility of amide protons. The residues that showed NOEs between the backbone amide proton and the N-bound methyls of the choline head group of DPC are shown in green. The amide residues that showed paramagnetic enhancement, ε, upon addition of 16-DSA are shown in magenta. The ε values are linearly coded on the basis of the intensity of magenta, with light pink corresponding to $ε = 0$ and dark magenta corresponding to $ε = ε_{max}$. **e** Solvent Accessible Surface Area (SASA, Å$^2$) from MD simulations of the Aβ(1-42) tetramer in DPC. Detergent micelle is represented as a smoothed transparent surface. The figure was prepared with the program Pymol. Source data are provided as a Source data file.

detected only when the spectrum was measured at the lowest pH. This observation suggested that residues comprising the β-turns and the N-termini ends exchanged faster with the solvent and were therefore more exposed than those making the β-sheet core and α1 (Fig. 2c). To establish whether the more protected β-sheet core residues exhibited distinct degrees of solvent protection, we determined their amide temperature coefficients (Δδ/ΔT). Most of the NH amide protons of residues comprising β1, β2, and β3 were the most affected by temperature changes, which is consistent with these residues forming stable hydrogen bonds[20]. In contrast, amide protons of β1 residues pointing out of the β-sheet core (i.e., Y10, V12, H14, K16, V18, and F20) exhibited the lowest amide temperature coefficients, suggesting that these residues are the most water accessible of all residues comprising the β-sheet core (Fig. 2c).

Next, to characterize the interaction of the DPC molecules with the surface of the Aβ(1-42) tetramer, we acquired a 3D $^{15}$N-resolved [$^1$H,$^1$H]-NOESY spectrum of the Aβ(1-42) tetramer using a selectively $^{13}$C methyl-protonated AILV and otherwise uniformly $^2$H,$^{15}$N Aβ(1-42) sample prepared using DPC at natural isotopic abundance. Analysis of this spectrum allowed us to identify two types of intermolecular interactions. First, we detected intermolecular NOEs between residues V12, L17, and L18, located in β1, and the N-bound methyl groups of the choline head group of DPC (Fig. 2d and Supplementary Fig. 10). Notably, this observation suggested that the detergent head groups bent towards the positively charged side chain of K16 located at the hydrophilic edges of the β-sheet core in order to stabilize them. Second, we detected intermolecular NOEs between all amide protons comprising the β-sheet core and the hydrophobic tail of DPC, with the largest intensities for residues located at the center of the β-sheet core and decreasing toward its edges (Fig. 2d and Supplementary Fig. 10). These observations were confirmed using a paramagnetic labeled detergent, 16-doxyl stearic acid (16-DSA) (Fig. 2d and Supplementary Figs. 11–13).

Finally, the interaction of the Aβ(1-42) tetramer with DPC micelles was further studied through molecular simulations using the SimShape approach[21]. Over the course of a 1 ns none-quilibrium simulation, the Aβ(1-42) tetramer was enveloped in a toroidal DPC micelle (Supplementary Fig. 14). Afterwards, the toroidal complex was equilibrated in explicit solvent for 60 ns.

**Aβ(1-42) tetramer—DPC interaction**. Having established the 3D structure of the Aβ(1-42) tetramer, we examined how it interacted with the surrounding media, namely water and the DPC detergent molecules. 2D [$^1$H,$^{15}$N]-HSQC spectra were acquired at pH 8.5 and 9.5 (Supplementary Fig. 9a, b). Residues belonging to the β-sheet core and some belonging to α1 were detected at both pHs, while some of the α1 residues and those corresponding to the β-turns and the N-termini ends were

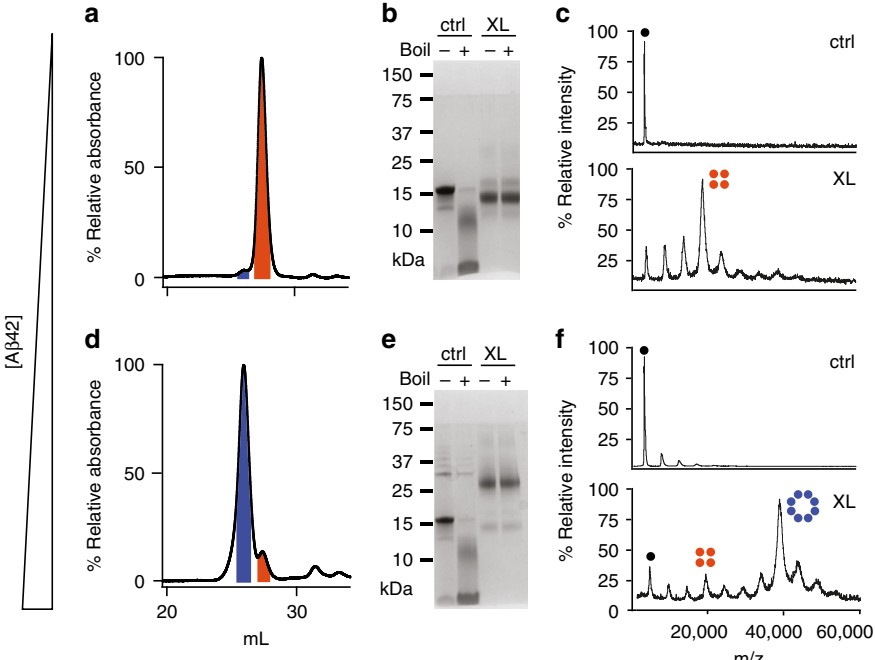

**Fig. 3 βPFO$_{Aβ(1-42)}$ samples can be enriched in either tetramers or octamers.** SEC of βPFOs$_{Aβ(1-42)}$ prepared at low (**a**) and high (**d**) Aβ(1-42) concentration in a column equilibrated in DPC. The peaks labeled in orange and blue are assigned, respectively, to Aβ(1-42) tetramers and octamers. SDS-PAGE analysis of βPFOs$_{Aβ(1-42)}$ prepared at low (**b**) and high (**e**) Aβ(1-42) concentrations either not cross-linked (ctrl) or having been previously cross-linked (XL) and showing the effect of boiling (+) and non-boiling (−). MALDI-TOF analysis of βPFOs$_{Aβ(1-42)}$ prepared at low (**c**) and high (**f**) Aβ(1-42) concentrations. Data shown in panels b and e are representative from three independent experiments. Source data are provided as a Source data file.

During this time, the hydrophobic terminal tail carbon of DPC was observed to interact predominantly with the two faces of the six-stranded β-sheet core, while transient contacts were also detected with the α1 region. Additionally, the DPC polar head was observed to interact with the hydrophilic edges of the six-stranded β-sheet region, which slowly became exposed to the solvent (Fig. 2e). Finally, these interactions were further validated by simulating the equilibrated protein-detergent complex in the absence of any external biasing forces (Supplementary Fig. 15). In summary, the experimental and the simulation results indicate that both faces of the central hydrophobic β-sheet core of the Aβ(1-42) tetramer were covered with a monolayer of DPC with α1 residues also interacting with the hydrophobic tail of DPC. In contrast, the rest of the residues, including the hydrophilic edges of the β-sheet core, were solvent-exposed and further stabilized by interactions with the polar head of DPC.

**βPFOs$_{Aβ(1-42)}$ sample contain Aβ(1-42) tetramers and octamers.** Previous electrical recordings using planar lipid bilayers had revealed that the βPFOs$_{Aβ(1-42)}$ sample induced various types of pore-like behavior[18]. Having established the 3D structure of the Aβ(1-42) tetramer, it was difficult to envision how it could be directly responsible for pore formation. For this reason, we attempted to determine whether other oligomer stoichiometries, not detectable by NMR, were present in the βPFOs$_{Aβ(1-42)}$ sample. To this end, we set to analyze the sample by means of size exclusion chromatography coupled to native ion mobility mass spectrometry (SEC/IM-MS)[22]. This strategy presented a unique opportunity to establish the stoichiometry of the potentially distinct oligomer species present as a function of their elution through a SEC column. We had previously analyzed βPFOs$_{Aβ(1-42)}$ in a SEC column equilibrated in DPC and shown that the sample eluted as a major peak at 27.4 mL (Fig. 3a). However, to carry out SEC/IM-MS, a different detergent that would be compatible with MS analysis and would preserve oligomer stability

was required[23]. C8E5 was found to fulfill both requirements (Fig. 4a). MS analysis of the early eluting volume of the βPFOs$_{Aβ(1-42)}$ peak revealed charge states consistent with the presence of tetramers and octamers. Analysis of the late eluting volume showed an increase in the relative abundance of the charge states corresponding to tetramers relative to those assigned to octamers. Importantly, the use of IM prior to MS analysis allowed unambiguous assignment of the contribution of distinct oligomer stoichiometries to each charge state (Supplementary Fig. 16 and Supplementary Table 3). This analysis led us to conclude that, although in agreement with NMR experiments, the stoichiometry of the major species present in the βPFOs$_{Aβ(1-42)}$ sample was Aβ(1-42) tetramers; Aβ(1-42) octamers were also present. In addition, since no charge states specific for other oligomer stoichiometries between tetramers and octamers were detected, these results suggested that tetramers were the building block for octamer formation. Notably, upon increasing activation conditions of the mass spectrometer, octamers did not decrease significantly in the spectrum at the maximum activation conditions afforded by the instrument (Supplementary Fig. 17), indicating that octamers were not derived from the forced co-habitation of two tetramers in a micelle but rather from specific interactions between the Aβ subunits composing it.

**Preparation of βPFOs$_{Aβ(1-42)}$ enriched in Aβ(1-42) octamers.** Having detected Aβ(1-42) octamers in the βPFOs$_{Aβ(1-42)}$ sample, we attempted to enrich our sample in this oligomer form to pursue its characterization. To this end, we maintained the concentration of DPC micelles constant and increased the concentration of Aβ(1-42) to mimic the consequences of an increase of the latter in the membrane[7]. Thus, from this point, we worked with two βPFOs$_{Aβ(1-42)}$ samples, one corresponding to the sample analyzed up to now and prepared at 150 μM of Aβ(1-42), referred to as βPFOs$_{LOW\_Aβ(1-42)}$, and one prepared at 450 μM Aβ(1-42), referred to as βPFOs$_{HIGH\_Aβ(1-42)}$. To establish whether

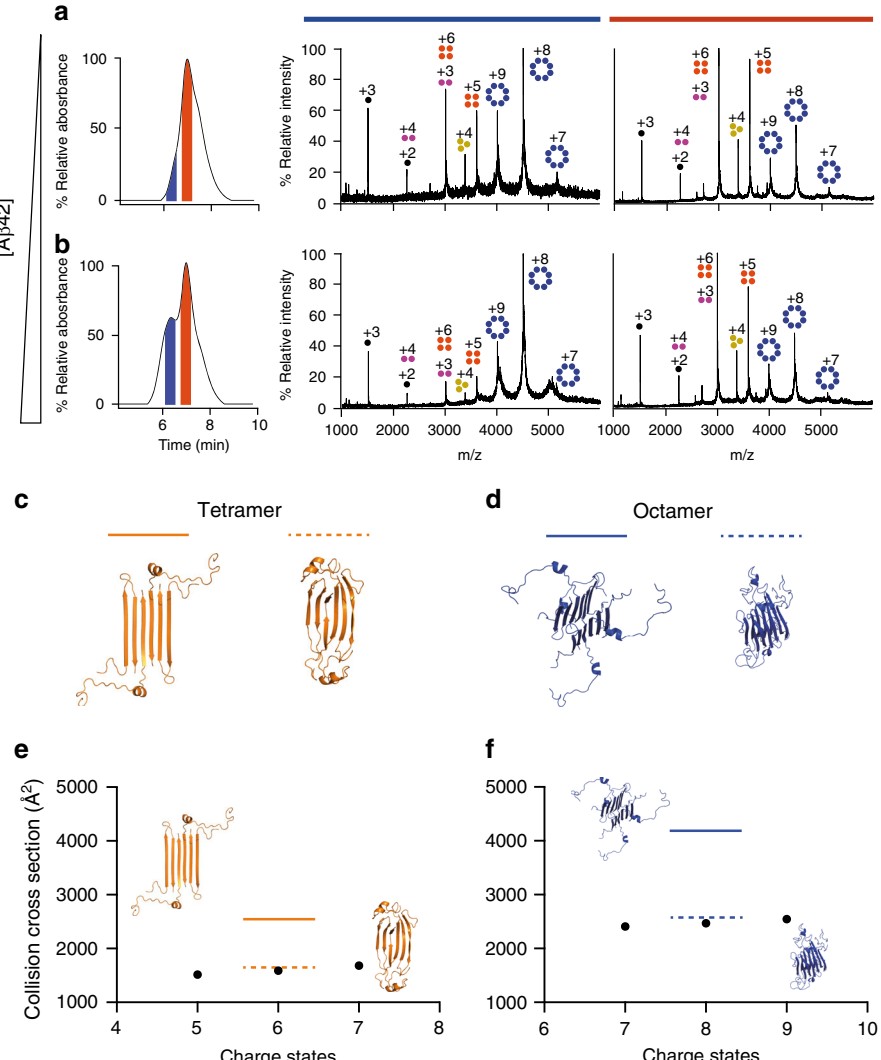

**Fig. 4 Aβ(1-42) octamers adopt a β-sandwich structure.** SEC-MS analysis of βPFOs$_{Aβ(1-42)}$ prepared at low (**a**) and high (**b**) Aβ(1-42) concentration. To couple SEC to MS analysis, the SEC column was equilibrated in C8E5. The mass spectra extracted from the blue and orange SEC peaks are shown, respectively, with a blue and orange line on top of them. The charge states corresponding to monomers, dimers, trimers, tetramers, and octamers are indicated with schematic drawings and labeled, respectively, in black, pink, yellow, orange and blue. **c** Aβ(1-42) tetramer structure derived from NMR restraints before (solid orange line) and after gas phase simulation for 100 ns prior to CCS calculation (dashed orange line). **d** β-sandwich octamer model based on the interaction of two Aβ(1-42) tetramers before (solid blue line) after gas phase simulation for 100 ns prior to CCS calculation (dashed blue line). **e** Experimental CCS of the tetramer (black dots) compared to the theoretical CCS of the Aβ(1-42) tetramer structure before (solid orange line) and after gas phase simulation for 100 ns prior to CCS calculation (dashed orange line). **f** Experimental CCS of the octamer (black dots) compared to the theoretical CCS of the β-sandwich structure before (solid blue line) and after gas phase simulation for 100 ns prior to CCS calculation (dashed blue line). Source data are provided as a Source data file.

βPFOs$_{HIGH\_Aβ(1-42)}$ were enriched in octameric forms, we analyzed them by SEC using a column equilibrated in DPC (Fig. 3d). This analysis resulted in a major peak eluting 1.4 mL earlier than βPFOs$_{LOW\_Aβ(1-42)}$, as well as a small peak eluting at the same volume as the major peak detected for βPFOs$_{LOW\_Aβ(1-42)}$. These findings indicated that working at high Aβ(1-42) concentration indeed led to the formation of a larger oligomer.

To study the stoichiometry of the oligomers present in the two samples, after preparing them in DPC micelles without any buffer exchange, we submitted them to chemical crosslinking. Given the abundance of basic and acid moieties in the flexible regions of the Aβ(1-42) tetramer structure derived by NMR (Supplementary Fig. 8c), we decided to generate zero-length (ZL) cross-links between Lys and Asp or Glu residues using DMTMM as a coupling reagent[24]. As previously described, SDS-PAGE analysis of the non-cross-linked βPFOs$_{LOW\_Aβ(1-42)}$ sample led, depending on whether

the sample had been previously boiled or not, to either a 5 kDa band, corresponding to Aβ(1-42) monomers, or to a major band at 18 kDa, consistent with Aβ(1-42) tetramers (Fig. 3b)[18]. In contrast, SDS-PAGE analysis of the cross-linked βPFOs$_{LOW\_Aβ(1-42)}$ sample led to a major band at 14 kDa, regardless of whether it had been boiled previously. The decrease in migration detected for the cross-linked samples is associated with protein compaction caused by crosslinking events[25]. To further confirm the stoichiometry of the cross-linked bands established by SDS-PAGE, samples were analyzed by MALDI-TOF (Fig. 3c). MALDI ionization involves harsh conditions, which prevents preservation of the non-covalent interactions present in protein complexes. Therefore, as expected, the molecular weight of the sample analyzed by MALDI-TOF without being cross-linked led to the detection of a peak corresponding to the molecular mass of the monomer (Supplementary Table 4). Instead, analysis of the cross-linked βPFOs$_{LOW\_Aβ(1-42)}$

sample led to the detection of a major peak consistent with the mass of an Aβ(1-42) tetramer, thereby confirming the suitability of the ZL chemistry to efficiently cross-link the major species formed under this condition. Next, we applied the same cross-linking chemistry to the analysis of the βPFOs$_{HIGH\_Aβ(1-42)}$ sample. SDS-PAGE analysis of the non-cross-linked samples led to the same bands obtained for βPFOs$_{LOW\_Aβ(1-42)}$, as well as to a faint band at about 30 kDa, consistent with Aβ(1-42) octamers (Fig. 3e). Instead, SDS-PAGE analysis of the cross-linked βPFOs$_{HIGH\_Aβ(1-42)}$ sample, both non-boiled and boiled, led to the detection of bands migrating at 28 kDa, consistent with Aβ(1-42) octamer formation. This result was further validated by MALDI-TOF analysis (Fig. 3f). All together, these results indicated that the βPFOs$_{HIGH\_Aβ(1-42)}$ sample comprises mainly Aβ(1-42) octamers. Moreover, the observation that SDS-PAGE analysis of the non-cross-linked and non-boiled βPFOs$_{LOW\_Aβ(1-42)}$ and βPFOs$_{HIGH\_Aβ(1-42)}$ samples led to mainly the same Aβ(1-42) tetramer band points to Aβ(1-42) octamers being formed by two tetrameric building blocks whose stabilizing interactions are not preserved in the presence of SDS.

**Aβ(1-42) octamers adopt a β-sandwich structure.** Subsequently, we analyzed βPFOs$_{HIGH\_Aβ(1-42)}$ by SEC/IM-MS. Although C8E5, the detergent required for native MS analysis, did not completely stabilize the larger oligomer detected in a SEC column equilibrated in DPC (compare Figs. 3d to 4b), analysis of the early eluting peak, corresponding to the larger oligomeric species, led almost exclusively to three charge states assigned to Aβ(1-42) octamers (Fig. 4b, Supplementary Figs. 18 and 19, and Supplementary Table 3). In summary, characterization of the βPFOs$_{LOW\_Aβ(1-42)}$ and βPFOs$_{HIGH\_Aβ(1-42)}$ samples by SEC, cross-linking/MALDI-TOF and SEC/IM-MS revealed that the former was enriched in Aβ(1-42) tetramers and the latter in octamers.

To study the conformational state of the Aβ(1-42) octamers, we used IM-MS to derive their collision cross-sections ($^{TW}CCS_{N2}$) (Supplementary Fig. 20 and Fig. 4c–f). We first validated this approach working with Aβ(1-42) tetramers, for which we had determined their 3D structure by NMR. Since a certain degree of compaction is expected in the gas phase[26], we simulated in vacuo the most representative charge state (+6) for the Aβ(1-42) tetramer structure for 100 ns. We observed a quasi-immediate and significant degree of compaction (gyration radius reduced by ~30%) that remained stable over the remaining 100 ns (Supplementary Fig. 21a). Compaction could be mainly attributed to the flexible N-termini ends while the β-sheet core remained stable throughout the simulation (Supplementary Fig. 21b). Next, we compared the experimental $^{TW}CCS_{N2}$ for the Aβ(1-42) tetramer (1598 Å$^2$) with the theoretical CCS obtained for the Aβ(1-42) tetramer structure determined by NMR after gas phase simulation (1647 Å$^2$) (Fig. 4c, e). This agreement between experimental and average CCS values after MD simulation validated the use of IM-MS to obtain insights into the structure of Aβ(1-42) octamers.

SDS-PAGE analysis of non-boiled samples enriched in Aβ(1-42) octamers led to a major band at 18 kDa as obtained for samples enriched in Aβ(1-42) tetramers (Fig. 3b, e). This result indicated that Aβ(1-42) octamers are derived from the assembly of two Aβ(1-42) tetramers. Consequently, to study the conformational state of the Aβ(1-42) octamers, we considered octamer models built from the assembly of two Aβ(1-42) tetramers. Considering the structure of the latter, we examined the association of two tetramers to either form a loose β-barrel or a β-sandwich structure. To this end, we simulated in vacuo for 100 ns the most representative charge state (+8) of both Aβ(1-42) octamer models. The behavior of the β-barrel and β-sandwich

structure along the simulation was very different (Supplementary Fig. 21). The β-sandwich structure showed a significant and quasi-immediate degree of compaction attributed to the flexible N-termini ends, as the β-sheet content remained stable to 40% throughout the simulation. Instead, the β-barrel structure compacted in three steps along the first 15 ns as a result of not only the flexible N-termini but also an immediate destabilization of its core β-sheets as shown by the ~20% decrease in β-sheet content. Since the theoretical CCS of the β-sandwich octamer (2546 Å$^2$) immediately matched the experimental $^{TW}CCS_{N2}$ for the Aβ(1-42) octamer (2469 Å$^2$) (Fig. 4f) and its compacted structure remained stable in vacuo, as observed for the Aβ(1-42) tetramers and as expected for membrane protein–micelle complexes in vacuo[27], we considered the β-sandwich structure the relevant topology for the Aβ(1-42) octamers. This result is indeed consistent with the physicochemical properties of the Aβ(1-42) tetramer as its two hydrophobic faces do not support its self-assembly in a β-barrel octamer structure with a central hydrophilic cavity. Instead, the Aβ(1-42) tetramer assembly in a β-sandwich octamer fully fulfils its physicochemical properties.

**Structures of βPFOs$_{Aβ(1-42)}$ reveal edge conductivity pores.** Having obtained the means to prepare and characterize βPFOs$_{Aβ(1-42)}$ samples enriched in tetramers and octamers, we set to compare their activity in lipid bilayers by electrical recordings using planar lipid bilayers (Supplementary Fig. 22). The only difference between the two samples was found in the occurrence rate of the different pore-like behaviors with βPFOs$_{LOW\_Aβ(1-42)}$, enriched in tetramers, exhibiting fast and noisy transitions with undefined open pore conductance values for a higher number of times than βPFOs$_{HIGH\_Aβ(1-42)}$, enriched in octamers, and the latter exhibiting a well-defined open pore with no current fluctuations for a higher number of times than the former. The observation of pore-like activity for Aβ(1-42) tetramer and octamer samples motivated the use of molecular dynamics (MD) simulations to probe the mechanism of bilayer disruption at an atomistic scale. These simulations involved the application of an external electric field to observe ion conductance properties in 150 mM NaCl, 310 K, at 100 mV for 500 ns.

We first monitored the conformational drift, structural flexibility, and secondary structural content of the Aβ(1-42) tetramer and octamer structures along the simulation time. RMSD of the Cα atoms of the tetramer and octamer structures revealed plateau levels of 2 and 2.5 Å, respectively (Supplementary Fig. 23). Analysis of the root mean square fluctuation (RMSF) showed the greatest fluctuations for the N-termini of both the red and green subunits and the loop region connecting β1 and β2 strands of the red subunit while almost no fluctuations were detected for the β1, β2, and β3 strands (Supplementary Fig. 24). Finally, analysis of β-sheet content of the membrane-bound Aβ(1-42) tetramer and octamer revealed that it remained stable along the course of the simulation (Supplementary Fig. 25). All together, these simulations and analyses indicated that the overall fold of the oligomers remained stable in a membrane bilayer environment with an applied electric field.

Next, we aimed at gaining insights into the mechanism by which Aβ(1-42) tetramer and octamer structures promoted bilayer disruption at an atomistic scale. The presence of hydrophilic residues on the edges of both the Aβ(1-42) tetramer and octamer structures resulted in their unfavorable exposure to the hydrophobic lipid tails of the membrane (Fig. 5). This situation led to lipid rearrangement, such that the head groups of the lipids reoriented to face the hydrophilic edges. Contacts between protein and DPPC head group atoms were characterized for both tetramer and octamer systems. This analysis revealed

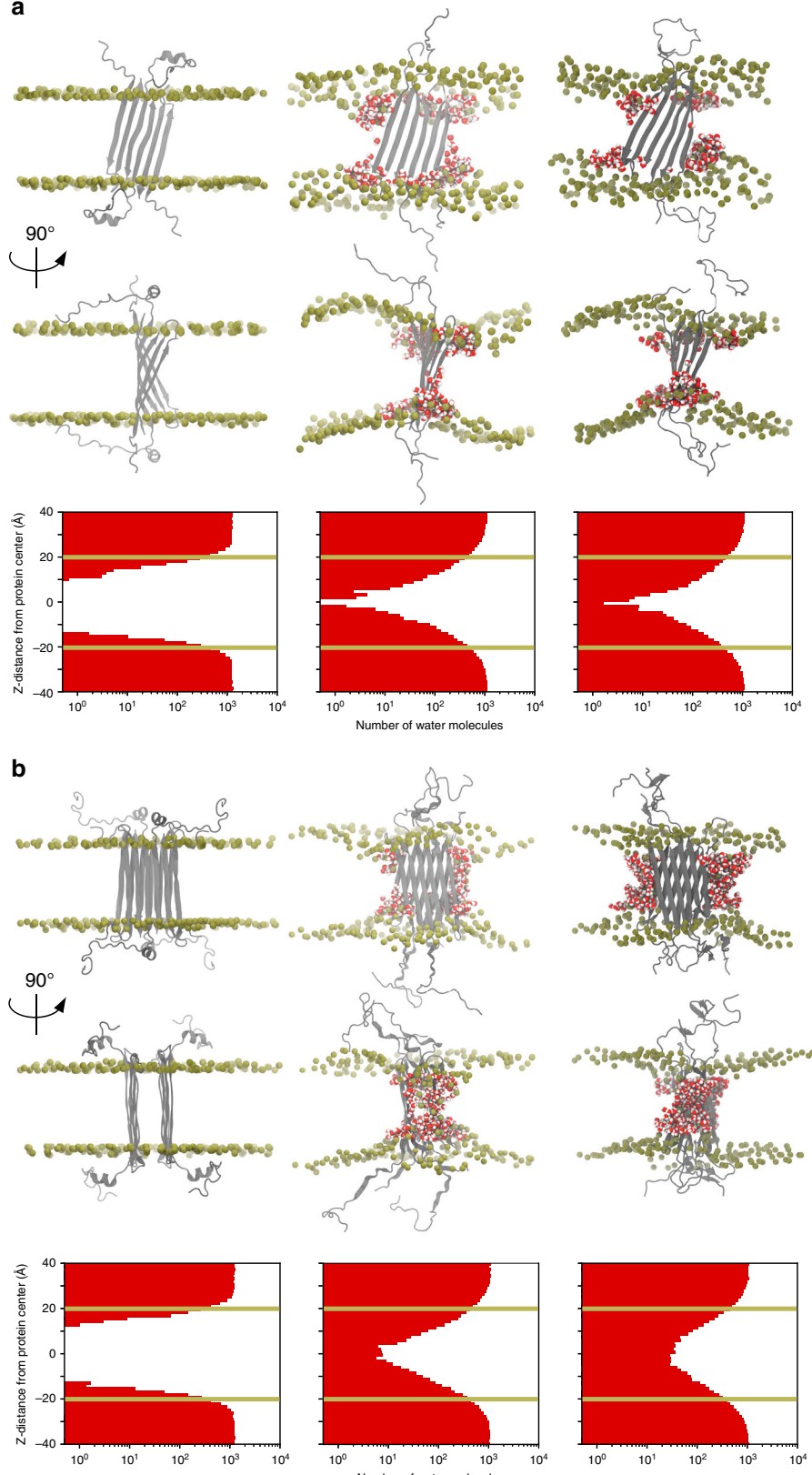

**Fig. 5 MD simulations in DPPC membrane bilayer of Aβ(1-42). a** tetramer and **b** octamer. The snapshots (two top rows) and water permeation profiles represented as histograms of the distribution of water molecules along the membrane normal (z) direction (bottom row) correspond to the initial coordinates (left), after 100 ns isothermal-isobaric NPT equilibrium simulation (middle), and after 500 ns canonical ensemble NVT simulation with 100 mV applied electric field (right). Protein is shown in grey, DPPC headgroup phosphorous atoms are shown in tan, and water in red/white. The tan lines in the water permeation profiles represent approximately the membrane spanning region.

that the headgroup of DPPC assembled towards the hydrophilic edges, specifically β1 strands, of the tetramers and octamers (Supplementary Figs. 26 and 27) leading to the formation of lipid-stabilized pores, which stabilized the protein–lipid complex. Subsequently, we analyzed water permeation profiles along the membrane normal (z) direction. We observed a higher degree of water permeation and a greater solvent-accessible surface area in the octamer than in the tetramer (Fig. 5). We associate the formation of lipid-stabilized pores observed during the MD simulations with the mechanism of water and ion permeation observed experimentally through electrical recordings using planar lipid bilayers (Supplementary Fig. 22) and propose them to explain the neurotoxicity observed in AD through the disruption of cellular ionic homeostasis.

## Discussion

In this study, we have used a multidisciplinary approach comprising the use of NMR, MS, electrophysiology, and MD simulations. This combined approach has allowed us to identify a putative Aβ form potentially responsible for AD neurotoxicity, as we have defined the structural and biophysical properties of membrane-associated Aβ(1-42) tetramers and octamers. To date, only the 3D structures of Aβ fibrils have been described[3–6] and no experimental structure has been reported for Aβ oligomers, only models. Compared to the structure of Aβ fibrils[3–6], the oligomers characterized in this study offer a 3D arrangement for Aβ(1-42) completely different: from the intermolecular formation of parallel β-sheets in Aβ fibrils to intramolecular and intermolecular antiparallel β-sheet formation in the membrane-associated Aβ(1-42) oligomers.

Compared to previously reported Aβ oligomer models, our NMR structure of Aβ(1-42) tetramers exhibits some similarities but also some differences. Thus, as in our tetramer structure, previous models of Aβ(1-42) peptides simulated in an implicit membrane[28] or revealed by solid-state NMR spectroscopy of an engineered variant that forms stable protofibrils[9] are based on a β-hairpin topology with residues D23 to K28 constituting the connecting loop. However, the topology of residues K16-A42, which in these models are organized in β-hairpins is different from the one presented in this study. While in our Aβ(1-42) tetramer structure residues G10-A21 and G29-V40 form two β-strands, β1 and β2, in the above described previous models, residues K16-G36 adopt a β-hairpin conformation and residues G39-I41 at the C-terminus form a third shorter β-strand following a turn involving residues G37-G38. Moreover, in these previous studies a single Aβ(1-42) conformation is considered in the simulations or is observed by solid-state NMR while in our study Aβ(1-42) adopts two distinct conformations in the tetramer structure. Finally, while Aβ(1-42) oligomers from solid-state NMR were modeled as circular hexamers[9], simulations of β-hairpin oligomers showed that Aβ(1-42) peptides could assemble as stable double-layered β-sheets with lateral association of β-hairpins in a parallel manner[28,29]. Similarly, our Aβ(1-42) octamers associate in double-layered β-sheets but with antiparallel association of β-hairpins. Therefore, our study widens the description of the much-needed low energy structural landscape of Aβ.

In addition, a strong link of this study to AD comes from the fact that the formation of the tetramers and octamers reported here is specific for Aβ(1-42), the variant most strongly linked to AD, versus Aβ(1-40), the variant most abundantly produced[18]. Having obtained the 3D structure for Aβ(1-42) tetramers allows to rationalize why its formation is specific for Aβ(1-42). Indeed, the absence of the two C(t) residues shortens the β-strands comprising the six-stranded β-sheet core from 14 residues to 12,

which is too short to span the hydrophobic portion of the lipid bilayer, thus preventing its stability in a membrane environment and providing a structural explanation to understand the different pathological role of Aβ(1-42) and Aβ(1-40) in AD.

One important implication of this study is to establish whether the results obtained in a micelle environment can be extrapolated to the lipid bilayer. To address this important point we will consider three scenarios: (i) insertion of Aβ(1-42) tetramers and octamers from a micelle to a lipid bilayer; (ii) formation of Aβ(1-42) tetramers and octamers within the lipid bilayer; and, (iii) stability of Aβ(1-42) tetramers and octamers in the lipid bilayer. The process of Aβ(1-42) tetramers and octamers insertion from a micelle to a lipid bilayer occurs in our electrophysiology experiments (Supplementary Fig. 22). The results from these experiments indicate that both samples enriched in Aβ(1-42) tetramers and octamers lead to specific pore-like behavior. In these experiments, assuming the structure observed in the micelle is maintained in the lipid bilayer, the N-termini of the Aβ(1-42) tetramers and octamers, with all their charged residues, are required to traverse through the hydrophobic core of the bilayer. This process, which is observed for the insertion of pre-assembled nanopores, such as ClyA and FraC nanopores into lipid bilayers, can be very efficient[30]. For example, entire proteins including GF, β-lactamase, organophosphorus hydrolase, and β-galactosidase can be fused to the N-terminus of a ClyA nanopore and transported across a lipid bilayer in vivo[31]. In the case of Aβ(1-42) tetramers and octamers, the unstructured nature of the charged N-termini would help traversing the bilayers as demonstrated for cationic cell-penetrating peptides[32]. In a cellular environment, we cannot rule out the possibility that the insertion of such structures is facilitated by other cellular components. However, systems such as the Sec machinery that aid membrane insertion usually require proteins with a signal peptide and they operate one monomer at the time. Hence, such a mechanism would not progress through oligomeric intermediates. Our results indicate that Aβ(1-42) tetramers and octamers can form pores in the hydrophobic environment of a lipid bilayer, but we cannot exclude that the complexity of biological membranes will affect the kinetics and thermodynamics of biological membrane insertion.

As per the formation of Aβ(1-42) tetramers and octamers in the native lipid bilayer, it is generally accepted that after APP cleavage Aβ is released from the membrane. However, several studies indicate that Aβ might accumulate in the membrane. Indeed, larger amounts of Aβ are detected in the membrane fraction than in the soluble one[7,33]. We therefore propose that Aβ accumulation in the membrane, as it occurs within micelles, could be the trigger for formation of Aβ(1-42) tetramers and octamers in the native environment of lipid bilayers. Finally, evidence for the stability of βPFOs$_{Aβ(1-42)}$ in a lipid environment come from a previous study where we showed structural stability in the lipid environment provided by bicelles[18]. Moreover, in this study we have performed long time-scale molecular dynamics (MD) simulations with applied electric field, which have revealed that both Aβ(1-42) tetramers and octamers maintain their overall fold.

Finally, apart from establishing the structure of membrane-associated Aβ(1-42) tetramers and octamers and assessing their pore-activity in planar lipid bilayers, MD simulations revealed that membrane disruption arises from the hydrophilic residues located on the edges of the β-sheets leading to the formation of lipid-stabilized pores. Such behavior resembles the toroidal pore-type behavior shown by many antimicrobial peptides[34] and would also be consistent with the reported antimicrobial activity for Aβ[35,36]. Moreover, the role of hydrophilic residues in membrane disruption is consistent with previous studies that show

small penalties to expose charged side chains, such as Lys and Arg, to lipids due to the stabilizing influence of membrane deformations for the protonated form[37,38].

In summary, we have established the 3D structure of an Aβ membrane-associated oligomer with the ability to form lipid-stabilized pores that could explain neurotoxicity in AD. We therefore present a unique opportunity to establish whether the Aβ(1-42) tetramers and octamers described in this study are indeed the Aβ species responsible for AD neurotoxicity. For example, by producing antibodies that specifically recognize them and subsequently using these antibodies to validate the Aβ(1-42) tetramer and octamer structures in AD brains. Therefore, the oligomers whose structure and function are described in this paper can be the long-sought Aβ species responsible for AD.

## Methods

**Reagents**. Lipids and detergents were purchased from Avanti Polar Lipids or Affymetrix-Anatrace. Deuterated reagents were purchased from Cortecnet or Eurisotop. All other reagents were supplied by Sigma-Aldrich unless otherwise stated. Kits for selective isotopically labeled samples were purchased from NMR-Bio. All buffers and solutions were freshly prepared using water provided by a Milli-Q system (18 MW cm$^{-1}$ at 25 °C, Millipore).

**Purification of synthetic Aβ samples and Aβ mixtures**. Aβ(1-42) and Aβ(17-42) were synthesized and purified by Dr. James I. Elliott (New Haven, CT, USA). The following protocol, described for Aβ(1-42) but applicable to the other Aβ peptides under study, was used to obtain Aβ in a monomeric state. In all, 5–10 mg of lyophilized synthetic Aβ(1-42) peptide were resuspended in 6.8 M guanidinium thiocyanate (GdnSCN) to a final concentration of 2.5 mg Aβ(1-42) mL$^{-1}$ and sonicated for 5 min in an ice bath. Afterwards, the sample was further diluted with Milli-Q water to 1.5 mg Aβ(1-42) mL$^{-1}$ and 4 M GdnSCN, and centrifuged. Finally, 2 mL of the 1.5 mg Aβ(1-42) mL$^{-1}$ solution was injected into a HiLoad Superdex 30 prep grade column (GE Healthcare), previously equilibrated with 50 mM ammonium carbonate. The fractions corresponding to monomeric Aβ(1-42) were collected and their purity and concentration were determined by Reversed Phase High Performance Liquid Chromatography (RP-HPLC). The pool was finally aliquoted in the desired amounts, freeze-dried, and kept at −20 °C until use.

For mixed samples containing different peptides such as Aβ(1-42)/Aβ(17-42), we purified, as described above, the most insoluble peptide first and prepared aliquots in the desired amounts, froze them with liquid nitrogen, and kept them at −20 °C. Afterwards the second peptide was purified in the same way, aliquots were added on top of the already frozen one and the combined aliquot was freeze-dried. Samples were kept at −20 °C until use.

**Expression and purification of recombinant Aβ samples**. The DNA encoding Aβ(1-42) was synthesized by PCR following KOD polymerase (Novagen) methods and using a modular approach[39], but with the following primers to add the 15 bp on each side for the In-Fusion method:

Fw 5′-GCGAACAGATCGGTGGTGATGCGG-A-GTTCCGTCATGATTCAG-3′ and

Rev 5′-ATGGTCTAGAAAGCTTTATTACG-CTATGACAACACCACCCACC ATGAG-TCCAATGATGGCACC-3′

The amplified DNA fragment was purified and cloned into a pOPINS vector[40] previously cut with KpnI and HindIII (New England Biolabs) restriction enzymes following the In-Fusion cloning method (Clontech). This resulted in a plasmid for the expression of Aβ(1-42) in the cytoplasm of Rosetta (DE3) pLysS E. coli cells (Novagen) as a fusion protein with an N-terminal hexahistidine SUMO affinity tag.

For all labeling schemes, Rosetta (DE3) pLysS E. coli cells (Novagen) were transformed with the expression vector and grown overnight at 37 °C on Luria Bertani (LB)-agar plates containing 1% glucose. All cell cultures were also supplemented with 35 μg mL$^{-1}$ chloramphenicol and 50 μg mL$^{-1}$ kanamycin. To enhance protein production, all Aβ peptides were expressed with the construct (His)$_6$-SUMO-Aβ using SUMO as a fusion partner. An auto-induction procedure was used to produce [U-$^{15}$N] Aβ(1-42) and [U-$^{15}$N] Aβ(17-42)[41]. Briefly, single colonies were picked and grown overnight in LB supplemented with 1% glucose. The pre-culture was centrifuged, and the pellet was transferred to $^{15}$N-labeled P-5052 auto-inducing media with the appropriate antibiotics. The resulting cultures were grown for 6 h at 37 °C. The temperature was then lowered to 25 °C, and the culture was incubated for a further 22 h. The cells were then harvested by centrifugation and frozen at −80 °C.

M9 minimal medium was used to produce [U-$^2$H,$^{13}$C,$^{15}$N] Aβ(1-42) and [U-$^2$H,$^{15}$N] Aβ(1-42) following previously reported protocols[41]. Briefly, single colonies were picked and grown overnight in LB supplemented with 1% glucose. Cells containing the DNA construct were adapted to grow in minimal medium in a stepwise manner by inoculating the cells into fresh M9 minimal medium containing increasing percentages of D$_2$O. The final pellet, already grown overnight

in M9 minimal medium prepared using 100% D$_2$O, was re-suspended and inoculated in 1 L M9 medium also prepared using 100% D$_2$O and containing 1 g L$^{-1}$ $^{15}$NH$_4$Cl and 2 g L$^{-1}$ D-glucose-$^{13}$C$_6$−1,2,3,4,5,6,6-d$_7$ or D-glucose-1,2,3,4,5,6,6-d$_7$. The culture was grown at 37 °C and induced at an OD$_{600}$ ~1 by the addition of IPTG to a final concentration of 0.5 mM. After overnight growth at 25 °C, the cells were harvested by centrifugation and then frozen at −80 °C.

For the production of selectively labeled Ile-[$^{13}$CH$^3$]$^{δ1}$, Ala-[$^{13}$CH$_3$], Leu/Val-[$^{13}$CH$_3$]$^{proR}$ Aβ(1-42) samples, we followed previously published procedures[42]. Briefly, 2-[$^{13}$CH$_3$], 4-[$^2$H$_3$] acetolactate (NMR-Bio) at 300 mg mL$^{-1}$ was added 1 h prior to induction. Forty minutes later (20 min prior to induction), 2-hydroxy-2-(1′-[$^2$H$_2$], 2′-[$^{13}$C])ethyl-3-keto-4-[$^2$H$_3$]butanoic acid (NMR-Bio) at 60 mg mL$^{-1}$ and 2-[$^2$H], 3-[$^{13}$C]alanine (NMR-Bio) at 700 mg mL$^{-1}$ were added. Protein expression was induced with IPTG.

The following protocol, described for Aβ(1-42) but applicable to the other Aβ peptides under study, was used to purify and obtain Aβ in a monomeric state. After protein expression, cells were lysed by sonication and centrifuged, and the supernatant was then purified as already described[41]. Briefly, the cleared soluble fraction was loaded onto a HisTrap HP 5-mL Ni column (GE Healthcare) and the fusion protein was eluted with 0.5 mM imidazole. IMAC fractions were analyzed by sodium dodecyl sulfate polyacrylamide gel electrophoresis (SDS-PAGE), and those containing the fusion protein were pooled. Next, the buffer was exchanged using a HiPrep 26/10 desalting column (GE Healthcare) equilibrated with 50 mM ammonium carbonate and 1 mM TCEP. Afterwards, the concentration and purity of protein was determined by Nanodrop® and RP-HPLC. Subsequently, samples were incubated overnight at 4 °C with SUMO protease (Ulp1) in a 1:50 protease: protein ratio to cleave Aβ(1-42) from the SUMO fusion tag. The concentration of Aβ(1-42) peptide after the cleavage was determined by RP-HPLC analysis. Subsequently, aliquots containing 3.75 mg Aβ(1-42) were prepared and freeze-dried. Each of these aliquots was solubilized with 6.8 M GdnSCN to 2.5 mg Aβ(1-42) mL$^{-1}$ and sonicated for 5 min in an ice bath. Afterwards, the sample was further diluted with Milli-Q water to 1.5 mg Aβ(1-42) mL$^{-1}$ and 4 M GdnSCN, and then centrifuged. Finally, 2.5 mL of the 1.5 mg Aβ(1-42) mL$^{-1}$ solution was injected into a HiLoad Superdex 30 prep grade column (GE Healthcare), previously equilibrated with 50 mM ammonium carbonate. The peaks corresponding to SUMO and monomeric Aβ(1-42) were collected separately, and their purity and concentration were determined by RP-HPLC. The pool containing pure Aβ(1-42) was aliquoted in the desired amounts, freeze-dried, and kept at −20 °C until use.

**Isotope-labeled samples for NMR experiments**. The following samples were produced: [U-$^{15}$N]-Aβ(1-42), [U-$^{15}$N]-Aβ(17-42) for 2D $^1$H-$^{15}$N-HSQC experiments; [U-$^2$H,$^{13}$C,$^{15}$N]-Aβ(1-42) for 3D backbone assignment experiments; [U-$^2$H,$^{13}$C,$^{15}$N]-Ile-[$^{13}$CH$^3$]$^{δ1}$, Ala-[$^{13}$CH$_3$], Leu/Val-[$^{13}$CH$_3$]$^{proR}$-Aβ(1-42) for side chain methyl assignments; and [U-$^2$H,$^{15}$N]-Ile-[$^{13}$CH$^3$]$^{δ1}$, Ala-[$^{13}$CH$_3$], Leu/Val-[$^{13}$CH$_3$]$^{proR}$-Aβ(1-42) for 3D $^{13}$CH$_3$-$^{13}$CH$_3$ and NH-$^{13}$CH$_3$ NOESY experiments.

**Preparation of βPFOs$_{Aβ(1-42)}$ sample for NMR experiments**. For NMR experiments, βPFOs$_{Aβ(1-42)}$ were prepared by dissolving lyophilized isotopically labeled monomeric Aβ(1-42) in the required volume of 10 mM Tris-d$_{11}$ and 28.5 mM DPC-d$_{38}$ to reach 1 mM Aβ(1-42). Afterwards, the pH was checked and adjusted to pH 9.5 or 8.5 with either a 10% HCl or a 10% NaOH solution, and the sample was left incubating at 37 °C for 24 h. To prepare βPFOs$_{Aβ42}$ at other Aβ(1-42) concentrations, only the concentration of DPC micelles ([M$_{DPC}$]) was adjusted so that the final [Aβ(1-42)]/[M$_{DPC}$] ratio was 2:1, where [M$_{DPC}$] is the concentration of DPC micelles and equals to the difference between the DPC detergent concentration ([D$_{DPC}$]) and its critical micellar concentration (CMC) divided by its aggregation number (i.e. ([D$_{DPC}$]-CMC)/aggregation number)). The CMC of DPC was taken to be 1.5 mM[10,43] and the DPC aggregation number 54[43]. This preparation, which is later referred to in the paper as βPFOs$_{LOW\_Aβ(1-42)}$, was found to be enriched mainly in Aβ(1-42) tetramers.

**NMR experiments**. All experiments were carried out at 37 °C on a 900 MHz Bruker Avance III HD spectrometer equipped with a 5-mm CP-TCI cryogenic probe, or an 800 MHz Bruker Avance III HD spectrometer equipped with a 3-mm CP-TCI cryogenic probe, both instruments located at the Swedish NMR Centre in Gothenburg, or an 800 MHz Bruker Avance III HD spectrometer equipped with a 5-mm CP-TCI cryogenic probe, located at IECB in Bordeaux.

For the resonance assignment of backbone atoms of the Aβ(1-42) tetramer prepared in DPC micelles, experiments from the standard Bruker library were recorded (HNCA, HNCACB, HNCO, and HN-NH NOESY). For the resonance assignment of methyl groups, experiments from the standard Bruker library were recorded (Hme)Cme([C]CA)CO, (Hme)Cme([C]CA)NH, Hme(Cme[C]CA)NH[44] and complemented with (H)C-TOCSY-C-TOCSY-(C)H experiments[45]. Additionally, four 3D SOFAST-NOESY-HMQC experiments[44] were recorded to obtain NOE correlation between methyl groups (Hm-HmCm and Cm-HmCm) and between methyl and amide protons (Hm-NH and Cm-HN) of the Aβ(1-42) tetramer in DPC micelles. The acquisition parameters for all the NMR experiments carried out are summarized in Supplementary Table 5. All these experiments were acquired using non-uniform sampling (NUS) using TopSpin 3.5, processed with MDDNMR 2.5 software[45,46], and analyzed using CCPNmr Analysis 2.4.2.

**NMR amide temperature coefficients**. Amide temperature coefficients of the Aβ (1-42) tetramer were determined by measuring the 2D [$^1$H,$^{15}$N]-TROSY spectra of the Aβ(1-42) tetramer sample at 303, 310, 317, and 324 K on a 900 MHz Bruker Avance III HD spectrometer equipped with a 5-mm CP-TCI cryogenic probe, and calculated using Eq. (1):

$$\text{Temperature coefficient values} = \Delta\delta_{NH}/\Delta T \qquad (1)$$

It is well established that, in aqueous solvents, exposed NHs typically display gradients from −6 to −8.5 ppb K1[47] while hydrogen-bonded exchange-protected NHs are characterized less negative Δδ/ΔT values than −4 ppb K1. However, numerous exceptions to these generalizations occur. In our case, the first anomalous observation was the fact that all amide protons presented positive Δδ/ΔT, meaning that chemical shifts of amide proton resonances shift downfield as the temperature increases. These downfield shifts with increased temperature may be explained by greater solvent protection of NH protons[47], probably due to the effect of the detergent micelle surrounding the tetramer. Furthermore, we observed that most of the NH amide protons of residues from β1, β2, and β3 were the most affected by temperature changes. Cierpicki et al. noted that amides involved in hydrogen bonds with a length of less than ~3.0 Å exhibited a larger temperature coefficient, because the secondary chemical shift caused by hydrogen bonding is greater, and so the same fractional change gives rise to a larger gradient[20]. This report would be consistent with residues with the largest Δδ/ΔT being involved in stable hydrogen bonds.

**NMR titrations with paramagnetic reagents**. NMR titrations with 16-DOXYL-stearic acid (16-DSA) were performed by addition of concentrated stock solutions of 16-DSA to an Aβ(1-42) tetramer NMR sample. Stock solutions of 16-DSA were obtained by dissolving this chemical in methanol-d$_4$. The Aβ(1-42) tetramer NMR sample was prepared at 1 mM Aβ(1-42), using the appropriately labeled sample, in 10 mM Tris-d$_{11}$, 28.5 mM DPC-d$_{38}$ at pH 9.5. 16-DSA stock solution was subsequently added to this Aβ(1-42) sample to obtain the following concentrations of 16-DSA: 0.3, 0.45, 0.6, 1.2, and 2.4 mM. [$^1$H,$^{15}$N]-TROSY and [$^1$H,$^{13}$C]-HMQC experiments were acquired at each 16-DSA concentration point. These NMR experiments were also performed on a sample without 16-DSA in order to be used as reference.

**NMR structure calculation of the Aβ(1-42) tetramer**. The structure of the Aβ(1-42) tetramer was determined with the iterative ARIA 2.3.2/CNS 1.21 software[48,49]. Distance restraints were derived from NOE cross-peaks (HN-HN, HN-Methyl, Methyl-Methyl) and used as input for ARIA 2.3.2, together with dihedral angle restraints and hydrogen-bond restraints. Upper bound distances for NOE restraints were derived from NOE cross-peaks volumes using characteristic distances (sequential NH-NH NOEs in β-sheet or intra-residual NH-Methyl in Alanines). Backbone dihedral angles were predicted from backbone chemical shifts with TALOS-N 4.21[50]. Predictions classified as strong were converted to dihedral angle restraints with an error corresponding to twice the standard deviation given by TALOS-N 4.21. Hydrogen bond restraints for anti-parallel beta-strand pairing were deduced from the NOE pattern and confirmed by initial calculations from NOEs and dihedral angle restraints only. Each hydrogen bond is encoded by two restraints (HN…O with upper-bound 2.3 Å and N…O, upper-bound 3.3 Å). During structure calculation, four copies of an Aβ(1-42) chain were modeled, using NCS restraints to maintain each dimer superimposable in the tetramer. For each iteration, 100 conformations were generated, except for the last iteration, where 500 conformers were calculated. The 50 lowest-energy conformers were refined in a shell of DMSO molecules[51] and the 15 refined conformers with the least number of distance restraint violations were selected as the final Aβ(1-42) tetramer ensemble. The structure ensemble was validated with PROCHECK 3.5.4[52] WHATIF 8.3[53] and MolProbity 3.19[54]. The coordinates for the Aβ(1-42) tetramer structure have been deposited in the Protein Data Bank with accession code 6RHY.

**Simulations of Aβ(1-42) tetramer solubilization in DPC**. From the NMR ensemble of Aβ(1-42) tetramer structures determined with ARIA, the conformer with the least restraint violations was selected to be solubilized in n-dodecylphosphocholine (DPC) micelle. The SimShape protocol with NAMD 2.13 was used to accelerate the assembly of the protein–micelle complex[21]. This method uses grid-steered molecular dynamics to assemble detergents into a toroidal micelle that wraps around the hydrophobic core of a membrane protein. Three biasing potentials were utilized in total. Two took the shape of concentric toroids, where the first toroid was coupled to the head-group heavy atoms of DPC, while the second smaller toroid was coupled to the tail-heavy atoms. The third biasing potential took the shape of a plane and was coupled to the detergent tail-heavy atoms.

Prior knowledge of the approximate micelle shape and orientation facilitated the design of the shape of the toroidal potentials. HDX studies characterized the central six-stranded β-sheet region of the tetramer with slow water exchange, suggesting burial of these residues within the micelle. This ~29 Å long region of the Aβ(1-42) tetramer was used to inform the dimensions of the toroid-shaped grid potentials. The number of DPC molecules in the micelle was determined to be 120 by estimating the size of the tetramer-micelle complex using overall correlation

time obtained from NMR[18]. The DPC molecule positions were initialized in a toroidal pattern around the protein such that there was a minimum distance of 10 Å between any given pair of detergent and protein atoms.

The complex was simulated using NAMD with grid-steered molecular dynamics for 1 ns at 310 K with a Langevin thermostat with a damping coefficient of 5 ps. Generalized Born Implicit Solvent (GBIS) with ionic strength 0.15 mM and dielectric constant of 80 were used. CHARMM36 forcefield parameters were used. Head and tail atoms were separately coupled to their grid potentials with scaling factors of 0.18 and 0.25, respectively. The tail atoms were additionally coupled to the planar grid potential with a scaling factor of 0.14. Backbone heavy atoms were harmonically restrained using a 1 kcal (mol Å$^2$)$^{-1}$ spring constant to maintain protein structure during detergent assembly. After 1 ns, detergent molecules assembled into a micelle around the protein with a toroidal shape similar to the attractive grid potentials (Supplementary Fig. 14).

Continuing with the SimShape protocol, the micellar complex was completely uncoupled from the toroidal attractive potentials, solvated with TIP3P water, and ionized with 150 mM NaCl. The solvated system was minimized with conjugate gradient energy minimization for 5000 steps, and simulated for 30 ns with protein backbone heavy atom harmonic position restraints with a spring constant of 1 kcal (mol Å$^2$)$^{-1}$. Next, the protein restraints were gradually removed to allow for complete equilibration of the complex (Fig. 2e). The spring constant was lowered in steps by 0.001 kcal (mol Å$^2$)$^{-1}$ every 10 ps for 10 ns. Finally, eight replicates of the unrestrained protein–micelle complex were simulated for an additional 100 ns each with no biasing forces applied. These equilibrium trajectories were then used to characterize contacts between the tetramer and detergents.

Contacts were calculated between the Aβ(1-42) tetramer and DPC molecules and are shown in Supplementary Fig. 15. Two contact sites on DPC were considered: the head group nitrogen atom and the terminal tail carbon atom. The number of contacts was defined as the number of contact sites within 9 Å of an amide backbone nitrogen atom and was calculated for every frame, summed over symmetric chains, and averaged over a 100-ns trajectory, returning an average number of contacts per residue. The per-residue average contact number of the eight independent replicates was then considered to be eight independent samples, allowing for calculation of statistical error per residue, shown as ±standard deviation divided by the number of independent samples.

**Preparation of βPFOs$_{LOW\_A\beta(1-42)}$ and βPFOs$_{HIGH\_A\beta(1-42)}$**. βPFOs$_{LOW\_A\beta(1-42)}$ and βPFOs$_{HIGH\_A\beta(1-42)}$ corresponding to Aβ(1-42]/[M$_{DPC}$] ratios of 2:1 and 6:1, respectively, were prepared from freeze-dried monomeric Aβ(1-42) samples and dissolved in 10 mM Tris, 5.5 mM DPC adjusted to pH 9, reaching a final concentration of 150 μM Aβ(1-42) in the case of βPFOs$_{LOW\_A\beta(1-42)}$ and 450 μM Aβ(1-42) in the case of βPFOs$_{HIGH\_A\beta(1-42)}$. The samples were incubated at 37 °C for 24 h.

**SEC**. βPFOs$_{LOW\_A\beta(1-42)}$ and βPFOs$_{HIGH\_A\beta(1-42)}$ samples were injected into a tandem Superdex 200 increase 10/300 (GE Healthcare). The columns were equilibrated with 10 mM Tris, 100 mM NaCl at pH 9 containing 3 mM DPC and eluted at 4 °C at a flow rate of 0.5 mL min$^{-1}$. Data was collected and analyzed using Unicorn v7.0 (GE Healthcare).

**SEC/IM-MS**. An Acquity UPLC H-class system (Waters, Manchester, UK) comprising a quaternary solvent manager, a sample manager set at 10 °C, a column oven and a TUV detector operating at 280 nm and 214 nm was coupled to a Synapt G2 HDMS mass spectrometer (Waters) for online SEC/IM-MS instrumentation[22]. An Acquity BEH SEC column (4.6 × 150 mm, 1.7 μm particle size, 200 Å pore size) (Waters) was equilibrated with 200 mM (NH$_4$)$_2$CO$_3$, 14.2 mM C$_8$E$_5$ at pH 9.0 and run with the following flow rate gradient: 0.25 mL min$^{-1}$ over 4 min; then 0.10 mL min$^{-1}$ over 6 min and finally 0.25 mL min$^{-1}$ over 2 min.

The IM-MS experiments are reported as recommended by Gabelica et al[55]. The Synapt G2 HDMS was operated in positive mode with a capillary voltage of 3.0 kV. The main parameters—sample cone 180 V, trap collision energy 100 V, trap gas flow 5 mL min$^{-1}$ and backing pressure 6 mbar—were finely tuned to disrupt the detergent protein interaction and to maintain the oligomer species. Acquisitions were performed in the m/z range 1000–10,000 with a 1.5-s scan time. External calibration was performed using singly charged ions produced by a 2 g L$^{-1}$ solution of cesium iodide dissolved in 2-propanol/water (50/50, v/v). To assess the effect of activation energy on the oligomeric species detected, we increased the energy conditions by means of the sampling cone (100, 180 and 200 V) and the trap collision energy (50, 100 and 160 V). MS data collection and analysis were performed using MassLynx 4.1 (Waters).

In order to unambiguously assign the oligomeric state of each MS peak and to measure the drift time of each species, an ion mobility method using travelling wave IMS (TWIMS) technology was optimized as described below. Prior to TWIMS separation, ions were thermalized in the helium cell (180 mL min$^{-1}$). Subsequently, ion separation was performed in the pressurized ion mobility cell using a constant N$_2$ (purity > 99%) flow rate of 90 mL min$^{-1}$. The wave height and velocity were 40 V and 800 m s$^{-1}$, respectively. Transfer collision energy was set to 15 V to extract the ions from the IM cell to the TOF analyzer. IM-MS experiments were performed in triplicate under identical instrumental conditions. Ion mobility

data was collected and analysed using DriftScope 2.4 (Waters). We assessed the CCS from the mobility measurements using Eq. (2)[56]:

$$\text{CCS} = \frac{3}{16}\sqrt{\frac{2\pi}{\mu k_B T}}\frac{ze}{NK} = \frac{3}{16}\sqrt{\frac{2\pi}{\mu k_B T}}\frac{ze}{N_0\left(\frac{p}{p_0}\frac{T_0}{T}\right)K} = \frac{3}{16}\sqrt{\frac{2\pi}{\mu k_B T}}\frac{ze}{N_0 K_0} \quad (2)$$

IM data were calibrated to perform CCS calculations using the most intense charge state of external calibrants prepared under non-denaturing conditions. The choice of calibrants is a critical point in the calibration framework[57]. The calibrants used were cytochrome C, β-lactoglobulin monomer and dimer, and avidin, which were in the same range in terms of MW, z, and CCS as those of the Aβ(1-42) tetramers and octamers used in this study (Supplementary Fig. 20). Theoretical CCS for our samples were derived from atomic coordinates of the NMR structure of the Aβ(1-42) tetramer and two octamer models built using the structure of the Aβ(1-42) tetramer as a building block. The first octamer model was based on the association of two tetramers to form a loose β-barrel structure and the second one on the association of two tetramers in a β-sandwich structure. Theoretical CCS value for the Aβ(1-42) tetramer and octamers were calculated using the previously described structures after 100 ns MD simulation in the gas phase (Supplementary Fig. 21). Gas phase simulations were prepared using the default vacuum simulation parameters in the QwikMD plugin of VMD 1.9.4a37, and extended to 100 ns. The QwikMD prepared gas phase simulations were ran using NAMD 2.13, with the CHARMM36 forcefield, using a 2.0 femtosecond time step, and at a temperature of 300 K. The two most abundant charges states for the tetramer (+6) and octamer (+8) were chosen as ionization states. The results depicted in Fig. 4 were obtained with a charge distribution considering that the oligomers were prepared at pH 9.0 and that during positive electrospray only residues not protected by the micelle got protonated. To establish that charge location was not biasing the experiments two additional charge distributions consistent with the tetramer (+6) and octamer (+8) were built and simulated in the gas phase for tetramer and octamers, for a total of 100 ns of gas phase simulations. Snapshots of the simulation trajectories were taken every 2 ps, and used to obtain theoretical CCS values using the Projection Approximation method within the Impact 1.0 software[58]. Final values for all three charge distributions were within 60 Å[2] of their mean values for CCS calculations of tetramers and octamers. Their corresponding PDB files were used as an input file and were run on Impact with a convergence value of 1% enabled to determine an average CCS as a mean of three independent calculations. Solvent and heteroatoms were excluded for the theoretical calculation.

**Cross-linking of βPFOS_LOW_Aβ(1-42) and βPFOS_HIGH_Aβ(1-42).** βPFOS_LOW_Aβ(1-42) and βPFOS_HIGH_Aβ(1-42) were prepared as described in the section Preparation of βPFOS_LOW_Aβ(1-42) and βPFOS_HIGH_Aβ(1-42), with the exception that 10 mM sodium carbonate (Na₂CO₃) was used instead of 10 mM Tris to avoid the interference of Tris in the cross-linking reaction. After sample preparation, the concentration of βPFOS_LOW_Aβ(1-42) was maintained at 150 μM Aβ(1-42) while that of βPFOS_HIGH_Aβ(1-42) was brought from 450 μM to 150 μM Aβ(1-42) by diluting it with a solution containing 10 mM Na₂CO₃, 1.5 mM DPC at pH 9. Afterwards, both samples were cross-linked using (4-(4,6-dimethoxy-1,3,5-triazin-2-yl)−4-methyl-morpholinium chloride) (DMTMM) as the cross-linking reagent[24]. DMTMM was added to a final concentration of 15 mM (4.15 mg mL⁻¹) and the samples were incubated for 2 h at 50 °C and 800 rpm. Samples were quenched by directly preparing them for SDS-PAGE and high-mass MALDI analysis.

**SDS-PAGE analysis of cross-linked oligomer samples.** The cross-linked samples were diluted to 50 μM Aβ(1-42) using a solution of 10 mM Na₂CO₃ and 1.5 mM DPC at pH 9. Finally, 20 μL of the resulting solution was mixed with 10 μL of 3X sample buffer (3X SB) and 20 μL of the mixture, either non-boiled or boiled (for 5 min at 95 °C) were electrophoresed in 1-mm thick SDS-PAGE gels containing 15% acrylamide. Gels were run at 50 V for 30 min, 120 V for 2 h and stained using Coomassie Blue.

**High-mass MALDI-MS analysis of cross-linked oligomer samples.** Before MALDI-TOF analysis, cross-linked samples were diluted down to 37.5 μM Aβ(1-42) in H₂O. This dilution step is critical to reduce the amount of detergent that could later interfere with the co-crystallization of the sample with the matrix, which is an essential prerequisite for the MALDI ionization process[59]. Next, diluted samples were mixed (1:1 v/v) with a matrix solution of sinapic acid (10 mg mL⁻¹) containing (1:1 v/v) acetonitrile/deionized water with 0.1% trifluoroacetic acid (TFA). Each mixture (2 μl thereof) was deposited on the MALDI target plate using the dried-droplet method. As a control, 2 μl of βPFOS_LOW_Aβ(1-42) and βPFOS_HIGH_Aβ(1-42) samples prepared as described but without adding the cross-linking reagent were examined using the same deposition method. High-mass MALDI-MS analyses were carried out on a MALDI-TOF mass spectrometer (Autoflex III, Bruker) used in linear mode and equipped with a HM3 high-mass detector (CovalX AG), which allows the (sub-μM) detection of macromolecules up to 1500 kDa with low saturation. Calibration was achieved using singly and doubly charged bovine serum albumin ions ([M + 2 H]²⁺= 33216 Da and [M + H]⁺= 66431 Da) and the gas phase dimer of this protein ([2 M + H]⁺= 132861 Da). The mass spectra were acquired by averaging 2000 shots (8 different positions into each spot and 250 shots per position), using the

same laser fluency before and after crosslinking. The spectra were processed (including background subtraction and smoothing) using FlexAnalysis 3.4.

**Electrical recordings with planar lipid bilayers.** Ionic currents from planar bilayers formed from diphytanoyl-sn-glycero-3-phosphocholine in 10 mM Tris·HCl and 150 mM NaCl at pH 7.5 and 23 °C were measured by applying a 2 kHz low-pass Bassel filter with a 10 kHz sampling rate. Potentials were applied, and the current was recorded using Ag/AgCl electrodes connected to a patch-clamp amplifier (Axopatch 200B, Axon Instruments). Current recordings were analyzed using the Clampfit 10 software package (Molecular devices). Open-pore currents were measured by a Gaussian fit to all-point histogram. The center of the peak corresponds to the open-pore conductance and the width at half height to the error. Each electrophysiology chamber contained 500 μL 10 mM Tris·HCl and 150 mM NaCl at pH 7.5. Two samples were analyzed, βPFOS_LOW_Aβ(1-42), which was diluted from 1:250 to 1:100 in the chamber[18], and βPFOS_HIGH_Aβ(1-42), which was diluted 1:100 in the chamber. For βPFOS_LOW_Aβ(1-42), type 1, 2, and 3 pores were observed in 17%, 48%, and 35% of the experiments (N = 105). For βPFOS_HIGH_Aβ(1-42), type 1, 2, and 3 pores were observed in 8.5%, 35%, and 56% of the experiments (N = 71). Controls were carried out to establish that the concentration of the detergent micelles present in the samples did not affect the stability of the bilayer.

**Aβ(1-42) tetramer and octamer simulations in a DPPC bilayer.** Molecular dynamics simulations of the Aβ(1-42) tetramer and octamer in planar lipid bilayers were performed under an applied electric field using NAMD 2.13. The protonation state of the titrable amino acids was chosen as that expected at pH 7.4 (lysine and arginine positively charged and aspartic and glutamic negatively charged). The ionization state of all histidine sidechains were set to neutral, using the HSD parameters of the CHARMM36m forcefield. Two structures were simulated, each in triplicate, one corresponding to the Aβ(1-42) tetramer structure obtained by NMR and the other to the octamer β-sandwich structure determined by CCS. These structures were aligned to the principal axes of their β-sheet regions, and embedded into 80Åx80Å planar 1,2-dipalmitoyl-sn-glycero-3-phosphocholine (DPPC) bilayers using the CHARMM-GUI input generator and CHARMM36m force field[60]. The systems were further simulated at equilibrium NPT conditions for 100 ns with semi-isotropic pressure coupling in the X–Y plane. The systems were then equilibrated in an NVT ensemble for 10 ns. Then an external electric field of 100 mV was applied along the Z-axis and each system was simulated for another 500 ns.

Secondary structure content for the membrane-bound Aβ(1-42) tetramer and octamer was also calculated for the simulations with applied electric field. For each system, the dictionary of protein secondary structure (DSSP) assignments were calculated at each frame using the MDtraj 1.9.3 python package[61]. The helix content was defined as the fraction of residues in α-helix conformation, while the β-sheet content was defined as the fraction of residues in a β-strand conformation. The secondary structure content for each replicate of the tetramer and octamer systems was obtained. For each system, the average β-sheet content of each replicate simulation was calculated, allowing for calculation of statistical error, which is shown as ±standard deviation between the three replicates (Supplementary Fig. 25).

Water permeation profiles were also calculated at three timepoints along the membrane-bound simulations: the initial (post-minimization, pre-equilibration) structure, after 100 ns equilibrium simulation, and after 500 ns simulation with applied electric field. These histograms represent the distribution of water along the membrane normal (z) direction. These water permeation profiles were then averaged over the three replicates of each system, and the average shown in Fig. 5.

Contacts between the bilayer DPPC molecules and the Aβ(1-42) tetramer and octamer during the applied 100 mV electric field were calculated. Two contact groups on DPPC were considered: the headgroup nitrogen and the two terminal tail carbon atoms. The number of contacts was defined as the number of contact sites within 9 Å of an amide backbone nitrogen atom and was calculated for every frame, summed over symmetric chains, and averaged over the last 100 ns of the 500 ns trajectory (Supplementary Figs. 26 and 27), returning an average number of contacts per residue. The per-residue average contact number of the three independent replicates was then considered three independent samples, allowing for calculation of statistical error per residue, shown as ±standard deviation divided by the number of independent samples.

**Reporting summary.** Further information on research design is available in the Nature Research Reporting Summary linked to this article.

## Data availability
Data supporting the findings of this manuscript are available from the corresponding author upon reasonable request. A reporting summary for this Article is available as a Supplementary Information file. The source data underlying Figs. 1a, b, 2c, d, 3a–f, 4a, b, e, f and Supplementary Tables S3 and S4 are provided as a Source data file.

## Code availability
Accession codes for deposited data: coordinates have been deposited in the Protein Data Bank under accession number PDB 6RHY, and chemical shifts have been deposited in the Biological Magnetic Resonance Bank under entry 34396.

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

## Acknowledgements

We acknowledge Montserrat Serra-Batiste, Martí Ninot-Pedrosa, Margarida Gairí and Jesús García for helpful discussions, sample preparation and NMR data acquisition at earlier stages of the project, James Tolchard for helpful discussions and building octamer models, Marta Vilaseca, Marina Gay, Carol V. Robinson and Michael Landreh for helpful discussions and MS data acquisition at earlier stages of the project, and Oscar Hernandez for help in calibration of CCS measurements. This study was supported by MINECO (SAF2015-68789), the Fondation Recherche Médicale (AJE20151234751) and the Counseil Régional d'Aquitaine Limousin Poitou-Charentes (1R30117-00007559) to N.C. The authors G.M and N.C. acknowledge funds from Fundació La Marató de TV3 (20140730). B.B research was supported by the INCEPTION project (ANR-16-CONV-0005). E.T. was supported by the National Institutes of Health (P41-GM104601 and R01-GM123455) and also acknowledges computing resources provided by Blue Waters at National Center for Supercomputing Applications, and Extreme Science and Engineering Discovery Environment XSEDE (grant MCA06N060). V.O. research was supported by the Swedish Research Council Formas (2015-04614). S. Cianferani research was supported by Agence Nationale de la Recherche and the French Proteomic Infrastructure (ANR-10-INBS-08-03). N.C., S. Cianferani, T.B. and E.P. acknowledge the support of COST Action (BM1403). E.P. was a PhD fellow funded by MINECO (FPI). T.B. was a PhD fellow funded by Institut de Recherche Servier.

## Author contributions

N.C. designed and coordinated the project. S.C. prepared all the NMR samples and analyzed all the NMR spectra. M.M. and V.O. acquired most of the NMR experiments required for this study. B.B. calculated the structure of the Aβ(1-42) tetramer from NMR restraints and contributed to analyze CCS measurements. E.P. prepared all the MS samples and analysed them by SEC and SDS-PAGE. E.P., T.B., and S.Cianferani acquired and analysed the samples by SEC/IM-MS. E.P. and S.Chaignepain acquired and analysed the samples by MALDI-TOF. G.M. and M.B. performed and analysed electrical recordings using lipid bilayers. E.T., M.Meigooni, A.S.A. and J. D. carried out and analysed gas phase and detergent and membrane simulations. N.C. wrote the paper with input and contributions from all authors.

## Competing interests

The authors declare no competing interests.
