## [Peer Review File · Nature Communications]

Peer Review File -

Reviewers' comments first round:

Reviewer #1 (Remarks to the Author):

Carulla and co-workers employed various biophysical techniques - NMR, SEC/IM-MS (size exclusion chromatography coupled with ion-mobility mass spectrometry), MALDI-MS, electrical recordings, as well as MD (molecular dynamics) simulations - to determine a structure of micelle-bound Abeta(1-42) tetramers and octamers, which in case of the tetramer was submitted to the Protein Data Bank.

This work is of high importance and novelty as in the amyloid field structural information on Abeta oligomers (in solution and membrane-bound) is currently still missing. Such information is of importance as Abeta oligomers are thought to play a key role in the development of Alzheimer's disease.

While the importance of the current work is without doubt, some concerns have to be addressed before the manuscript may become publishable:

1) The authors need to mention that pH values between 8.5 and 9.5 are anything but physiological. It is known that the structures of monomeric and aggregated Abeta is highly susceptible to pH. Thus, the structures reported here might be of no relevance at all at physiological pH. Though having said this, it should also be mentioned that others publish Abeta fibril structures determined at a non-physiological pH of 2 in a solution involving 30% acetonitril (doi: 10.1126/science.aao2825). Thus, when such a fibril structure can be published in Science, it should also be warranted to publish an amyloid oligomer structure at pH 9 in Nature Commun. But the pH issue must be openly discussed.

2) When discussing the pH problem, the most likely protonation states of the amino acids at pH 9 should be given. For sure, histidines will be neutral (unlike what the authors state on pp. 7 and 8). But also the lysines may become neutral at such high pH, also considering that a hydrophobic environment reduces the pKa auf lysine. In the Methods regarding the MD simulations the chosen protonation states of titratable amino acids need to be given.

3) It should be further stated that the current structures determined from the co-assembly of Abeta and DPC into micelles containing Abeta are not necessarily those which one might find in lipid bilayers (biomembranes). With the suggested structure (Fig. 1d), the N-terminus of two of the peptides, with all its charged residues, would need to traverse through the hydrophobic core of the bilayer. The authors need to state how likely this event would be, e.g., how much energy it would take for this process to happen, or whether this could be enabled by transporters.

4) The MD simulations do not fully support that the suggested structures would be likely in a lipid bilayer environment. Fig. S14 shows that DPC assembles with its headgroup region around beta1, i.e., this strand prefers a polar environment and therefore draws water into the bilayer as Fig. 5 shows. As the simulations were only conducted for 100 ns, which is too short for a system of that size and needs to be extended to 1 microsecond or longer, it remains open how stable the suggested transmembrane tetramer structure indeed is. More analysis of the MD simulations needs to be done: RMSD/F, number of water molecules traversed, secondary structure per residue are some quantities that would be helpful.

5) While it is always expected that simulation scientists compare their results to existing experimental observations, it seems that the opposite is usually not expected. But almost 10 years ago models for transmembrane Abeta oligomers were published and their stability tested in MD simulations (doi: 10.1021/ja103725c; doi: 10.1016/j.bbamem.2012.09.001). Interestingly, later a very similar structure per Abeta peptide was determined by solid-state NMR as building block for Abeta hexamers: doi: 10.1002/anie.201406357. The authors should include a comparison of their structure model of membrane-bound Abeta oligomers with those previously published.

6) The CCS results in Fig. 4e and f are anything but convincing. Cutting off the flexible N-terminal ends to obtain agreement between structure model and actual CCS results does not increase the trust in the suggested tetramer and octamer models. However, it is anyhow questionable whether the ionized oligomers in the gas phase are structurally similar to the Aβ structures in DPC micelles. The authors should do more to convince this reviewer of this correspondence. As some of the authors are simulation scientists, one approach could be to simulate the oligomers in the proper ionization state in the gas phase to see how they evolve. The current approach (ignoring some of the residues to obtain the experimental CCS results) is below the standard expected for a publication in Nat. Commun.

Reviewer #2 (Remarks to the Author):

The manuscript by Ciudad et al. describes the structural characterization of Aβ(1-42) tetrameric and octameric structures. The structure of the tetramer is derived from NMR measurements. Cross-linking and SEC/IM-MS corroborate the findings of an Aβ tetramer, octamer, and other species. The study brings a 3D structure of an Aβ tetramer to the fore.

The introduction is composed of only two paragraphs and lacking background on Aβ or other oligomers that have been either modelled or determined experimentally. In short, the introduction would benefit from more background and context.

Aβ oligomers were studied in dodecylphosphocholine (DPC), a detergent, and at non-physiological pH (8.5 and 9.5). The authors note in discussion that DPC is a membrane mimic and not a physiological environment. However, there is no mention in regards to pH. Would the proposed structures be stable at physiological pH?

The 3D structure reveals an antiparallel arrangement with two of the three subunits exhibiting long flexible, solvent exposed N-termini. Given the structure, it would appear there would be large entropic barrier to insertion into the bilayer. Can the authors speculate on a plausible mechanism for insertion into the bilayer? Another issue not addressed, is the fact that charged and polar residues (such as His, Qln, and Lys) in their model (see Fig 1 and S17) are centrally located in the hydrophobic core of the bilayer. This simply does not make chemical sense and raises concerns to the validity of the proposed model. An explanation to rationalize exposure of such residues in the hydrophobic environment is warranted.

In regards to the SEC/IM-MS measurements, increased activation conditions were used to disrupt the Aβ complexes. The authors note tetramers broke into trimers and monomers, but this is not evident from the data presented (Fig S16 and S19). The assignment of oligomeric states by m/z and IM measurements is solid. The authors may consider isolating particular Aβ species using the quadrupole prior to subjecting them to increased activation? This has been done before (see doi: 10.1038/nature20820 and doi: 10.1007/s13361-016-1555-1), which would provide a more robust evaluation of complex gas-phase stability/dissociation.

IM measurements (CCS values) were compared to a limited set of Aβ models. The CCS values indicate a compact CCS value that agrees with models where the flexible loops have been removed from the structure prior to calculation. This approach raises significant concerns; deleting residues to match measured CCS values is not acceptable nor rigorous. First, flexible regions can lead to an increase in CCS values (see DOI: 10.1002/anie.201203047), which is not observed here. Does this imply that the N-termini are more ordered than flexible? Second, have other models been compared to the measured CCS values? Such as, a six-stranded cylindrin-like structure? Or other models? At present, additional models should be explored and also considering gas-phase minimization prior to CCS calculation.

No controls for the electrical recordings using planar lipid bilayers are presented. DPC is a detergent, which could disrupt the integrity of the bilayer giving rise to background signals. Were measurements of DPC in the absence of Aβ performed?

The section titled "preparation of ... enriched in Ab(1-42) octamers" seems out of place. Suggest moving up and before the SEC-MS analysis section.

The manuscript could be improved by merging some figures. For example, Fig S17 can be incorporated into Fig S7.

For denatured and native MS measurements, the measured and theoretical masses should be reported along with error.

Supplementary Fig. 18 is mislabelled. I presume the data shown is for PFOsHIGH not PFOsLOW.

Pg 8, "... using DPC at natural abundance." This is unclear as DPC is non-natural.

Pg 15, ".. we have precisely defined the structural .." Remove precisely.

Although I am not able to expertly comment on the quality of the NMR data, the study seems well designed and executed.

Reviewer #3 (Remarks to the Author):

This is a very interesting paper that presents a startling structure for the micelle-inserted amyloid-beta-42 polypeptide (AB42). AB42 is, of course, generally thought to be associated with the etiology of Alzheimer's disease. There is a body of evidence in the literature that AB42 can adopt some sort of transmembrane oligomer, with additional evidence that it can act (aberrantly) as an ion channel, potentially a mechanism that contributes to how AB42 promotes Alzheimer's disease. In this paper the authors used NMR spectroscopy to document that AB42 in dodecylphosphocholine micelles can form two closely related oligomeric structures. In one structure 4 subunits come together to form a single sheet with two subunits forming trans-micelle beta hairpins flank the two other subunits central in the tetrameric sheet as a single beta strands. What is amazing is that the 6-stranded tetrameric sheet does not form a closed barrel, but instead is an "open" sheet with solvent exposed edges for strands 1 and 6. The other structure is a dimer of tetramers where the 6-stranded sheets form a sandwich in the micelle interior, again with exposed edges. The authors also present electrophysiology measurements in which ion channel activity is documented, apparently for both monomer and dimers, with the channel activity apparently being associated with the exposed edges of the monomeric or dimeric sheets.

The NMR studies of this work were meticulously carried out and documented-- very convincing. Moreover, one can imagine how these structures might be stable in micelles: the horizontal diameter of the sheet is on the same order as the diameter of the micelle it sits in, so the need to form transmicelle beta barrels is relieved by the fact that the exposed edges of strands 1 and 6 are not exposed to a bilayer interior, but instead to water outside of the micelle water at the micelle-water interface. So, these structures seem reasonable as micelle structures. However, it seems less clear that they could be energetically accommodated as transmembrane structures in a sealed lipid bilayer, where the exposed edges of strands 1 and 6 would be exposed only to lipid, resulting in burial backbone amide groups with unsatisfied H-bonding potential.

The text, figures, and supporting material are all very well written. I think this work is EXTREMELY interesting and should be published following revision, without any requirement for additional experiments. However, the authors should provide more detail on what evidence they have that the micellar structure they carefully document in this work is maintained in lipid vesicles. This closely relates to the question of whether the channel activity they document really stems from AB42 channels that resemble the micellar structures. Again, more textual justification seems needed.

Finally, the sample composition, pH, and temperature need to be spelled out in detail in the caption to Figure 1.

Reviewer #4 (Remarks to the Author):

Review of NCOMMS-19-35967-T

The manuscript by Ciudad et al deals with an important open question in our understanding of the Alzheimer's Disease (AD) Pathology cascade: the mechanism by which amyloid-beta causes damages neurons. The authors cite the "amyloid hypothesis" in which Abeta is assumed to be the primary toxic agent in AD, and note that the mechanism of toxicity is unclear. They also cite works in which soluble forms of Abeta can induce pores of various sorts in membranes. This is the background for their basic hypothesis: that Abeta structures can form pores in planar lipid bilayers, and therefore can serve as a mechanism of neurotoxicity. The primary focus of the work is in establishing the feasibility of this idea, and the authors perform detailed work in establishing this to be the case. The evidence that Abeta structures can form pores in planar lipid bilayers, and that this is specific to certain forms of Abeta, seems to be well established by this study, and supported by beautiful and clear figures.

I will address a different aspect of this study, which is the validity of their hypothesis: that Abeta can cause pores in the membranes of neurons adjacent to sufficient concentrations of Abeta, which will then disrupt neuronal activity. The principal evidence for that disruption is shown in Supplementary Figure 20, in which the membrane conductance is increased by the introduction of β FOsA β (1-42) into planar lipid bilayers.

This makes a clear prediction of the effect of soluble Abeta on neuronal activity. The prediction is that the input conductance of these neurons will be increased. Intracellular recordings from several lines of APP+ transgenic mice (two of which are published), in which both soluble Abeta and amyloid plaques are present in the surrounding neuropil, by measurement of average input resistance in the neurons, do not show the increase in RIN that would be expected to be associated with such a conductance increase (Stern et al, J. Neurosci 24, 4335-4340 (2004); Kellner et al, Neurobiology of Aging 35, 1982-1991 (2014)). I do not know of any studies which have shown different results. (In addition, measurement by several groups show hyperactivity in numerous lines of APP transgenic mice, which is the opposite effect that would be expected from a general conductance increase.) these are issues that should be addressed by the authors, if they wish to argue that increased membrane conductance is indeed a mechanism of neuronal disruption by Abeta.

NOTE: This, however, is not necessarily evidence that the conductance increase does not occur. The direct measurements of input resistance are made using electrodes inserted into the soma. It may well be that the toxic effects of Abeta on the membrane are local. Indeed, one study shows that there are specific effects on dendritic spines in proximity to plaques, where the concentrations of soluble Abeta are higher (Spires et al, J Neurosci. 25, 7278-7287 (2005)).

This is an hypothesis which can be tested, using targeted patch recordings in slices of APP transgenic mouse cortex. The recordings would have to target dendrites in mice in which plaque aggregation is severe, so that dendrites would have a high probability of proximity to plaques. While difficult, targeted recordings are performed on a daily basis by numerous laboratories. If the input conductance is increased in these dendrites, compared to aged-match recordings in WT mice, it would indeed be strong evidence that the hypothesis proposed by the authors is correct. In many studies, including the one I cited, there is evidence for stripping of spines from these dendrites, which I would predict will show the opposite effect: raising the local dendritic input resistance.

At present, however, there is no such evidence of which I am aware that the mechanism proposed by the authors disrupts neuronal activity in AD. And some evidence does exist that the overall conductance in the neuron is not raised in APP-transgenic mouse model neurons. The current study does show an interesting possible mechanism of action of one of the forms of Abeta: one interesting enough to be further investigated. Again, evidence for the specific mechanisms (note the plural) of the pathophysiological effects of Abeta is a "missing link" in our understanding of the cascade of effects of the Abeta in the disease process.

Edward A. Stern
Brain Research Center
Bar Ilan University

Response to Reviewers Comments, first round -

Barcelona, 18 March 2020

We are delighted that all reviewers find our work interesting and are pleased to have the opportunity to address their points. We have now fully addressed the comments of reviewers 1, 2, and 3.

Reviewer 1

Carulla and co-workers employed various biophysical techniques - NMR, SEC/IM-MS (size exclusion chromatography coupled with ion-mobility mass spectrometry), MALDI-MS, electrical recordings, as well as MD (molecular dynamics) simulations - to determine a structure of micelle-bound Abeta(1-42) tetramers and octamers, which in case of the tetramer was submitted to the Protein Data Bank.

This work is of high importance and novelty as in the amyloid field structural information on Abeta oligomers (in solution and membrane-bound) is currently still missing. Such information is of importance as Abeta oligomers are thought to play a key role in the development of Alzheimer's disease. While the importance of the current work is without doubt, some concerns have to be addressed before the manuscript may become publishable:

Point 1. The authors need to mention that pH values between 8.5 and 9.5 are anything but physiological. It is known that the structures of monomeric and aggregated Abeta are highly susceptible to pH. Thus, the structures reported here might be of no relevance at all at physiological pH. Though having said this, it should also be mentioned that others publish Abeta fibril structures determined at a non-physiological pH of 2 in a solution involving 30% acetonitril (doi: 10.1126/science.aao2825). Thus, when such a fibril structure can be published in Science, it should also be warranted to publish an amyloid

oligomer structure at pH 9 in Nature Commun. But the pH issue must be openly discussed.

We appreciate this comment as it gives us the opportunity to better explain the reasons for working at pH 8.5 and pH 9.0. This paper is preceded by earlier work [Serra-Batiste, M. et al, Proc. Natl. Acad. Sci. USA 113 (2016)], where we first reported conditions to prepare β PFOsA β (1-42). As we aimed at characterizing physiologically relevant A β (1-42) oligomers, we carried out all screening conditions at pH 7.4, so conditions to prepare β PFOsA β (1-42) were established working at pH 7.4. However, we also found that β PFOsA β (1-42) adopted the same structure when prepared at pH 9.0 while being most stable. Since structural characterization of β PFOsA β (1-42) was facilitated when working with stable samples, we decided to work at pH 9.0. We have now added an explanation for the choice of pH in page 6 of the manuscript and added Supplementary Fig. 1. This figure shows that β PFOsA β (1-42) adopt the same structure when prepared at pH 7.4 and at pH 9.0, while they are most stable when prepared at pH 9.0.

Point 2. When discussing the pH problem, the most likely protonation states of the amino acids at pH 9 should be given. For sure, histidines will be neutral (unlike what the authors state on pp. 7 and 8). But also the lysines may become neutral at such high pH, also considering that a hydrophobic environment reduces the pKa of lysine. In the Methods regarding the MD simulations the chosen protonation states of titratable amino acids need to be given.

We agree with the reviewer that histidines are neutral at pH 9.0 and have consequently changed the sentence: “Notably, this observation suggested that the detergent head group is bent towards the positively charged side chains (i.e., H13, H14, and K16) located at the hydrophilic edges of the β -sheet core in order to stabilize them” by “Notably, this observation suggested that the detergent head groups bent towards the positively charged side chain of K16 located at the hydrophilic edges

of the β -sheet core in order to stabilize them.” (page 9). Also, in the methods section entitled “Simulations of the A β (1-42) tetramer and octamer in a DPPC bilayer”, we specify the protonation state of the titrable amino acids at pH 7.4, the pH at which the simulations have been carried out (page 53).

Point 3. It should be further stated that the current structures determined from the co-assembly of A β and DPC into micelles containing A β are not necessarily those which one might find in lipid bilayers (biomembranes). With the suggested structure (Fig. 1d), the N-terminus of two of the peptides, with all its charged residues, would need to traverse through the hydrophobic core of the bilayer. The authors need to state how likely this event would be, e.g., how much energy it would take for this process to happen, or whether this could be enabled by transporters.

We have now rewritten the discussion and wrote a paragraph describing the potential mechanisms for insertion of A β (1-42) tetramers and octamers from a micelle to a lipid bilayer (pages 19-20).

Point 4. The MD simulations do not fully support that the suggested structures would be likely in a lipid bilayer environment. Fig. S14 shows that DPC assembles with its headgroup region around beta1, i.e., this strand prefers a polar environment and therefore draws water into the bilayer as Fig. 5 shows. As the simulations were only conducted for 100 ns, which is too short for a system of that size and needs to be extended to 1 microsecond or longer, it remains open how stable the suggested transmembrane tetramer structure indeed is. More analysis of the MD simulations needs to be done: RMSD/F, number of water molecules traversed, secondary structure per residue are some quantities that would be helpful.

We appreciate this comment as it has allowed us to strengthen the conclusions of our

work. We have now extended the simulations to 500 ns and carried out further analysis: conformational drift by means of RMSD (Supplementary Fig 23), structural flexibility through RMSF (Supplementary Fig 24), β -sheet content along the simulation time (Supplementary Fig 25), and water permeation profiles along the membrane normal (z) direction (Fig. 5). All together, these analyses indicated that the overall fold of the oligomers remained stable along the simulation time and that water permeation occurred across the membrane space through the hydrophilic edges of the oligomers (page 16).

Point 5. While it is always expected that simulation scientists compare their results to existing experimental observations, it seems that the opposite is usually not expected. But almost 10 years ago models for transmembrane Abeta oligomers were published and their stability tested in MD simulations (doi: 10.1021/ja103725c; doi: 10.1016/j.bbamem.2012.09.001). Interestingly, later a very similar structure per Abeta peptide was determined by solid-state NMR as building block for Abeta hexamers: doi: 10.1002/anie.201406357. The authors should include a comparison of their structure model of membrane-bound Abeta oligomers with those previously published.

We have now rewritten the discussion and included a paragraph comparing the structural features of previously described A β oligomers models to those reported in this paper (page 18).

Point 6. The CCS results in Fig. 4e and f are anything but convincing. Cutting off the flexible N-terminal ends to obtain agreement between structure model and actual CCS results does not increase the trust in the suggested tetramer and octamer models. However, it is anyhow questionable whether the ionized oligomers in the gas phase are structurally similar to the Abeta structures in DPC micelles. The authors should do more to convince this reviewer of this correspondence.

As some of the authors are simulation scientists, one approach could be to simulate the oligomers in the proper ionization state in the gas phase to see how they evolve. The current approach (ignoring some of the residues to obtain the experimental CCS results) is below the standard expected for a publication in Nat. Commun.

We have followed the advice of the reviewer and carried out gas phase simulations of the structures of A β (1-42) tetramers derived from NMR restraints and that of two octamers models (β -barrel and β -sandwich) to obtain theoretical CCS values. The ionization states used in the simulations are described in the Methods section entitled "SEC/IM-MS" (pages 49-50) and the results of the gas phase simulations and the correspondence between experimental TWCCSN2 and theoretical CCS is now described in pages 14-15 and Figure 4.

An important point from obtaining theoretical CCS values from gas phase simulations is that it allowed us to revisit the calibrants used to obtain experimental TWCCSN2. Although a significant degree of compaction was observed after the simulations, the theoretical CCS values obtained did not reach the degree of compaction suggested by experimental ion mobility mass spectrometry (IM-MS) data. We had initially based our calibration on the work reported by Allison et al. [Allison, T.M. et al, Anal. Chem. 88 (2016)]. In this paper, they propose the use of soluble proteins as calibrants that are 30% larger in molecular weight (MW) than the membrane protein under study.

The higher MW of these soluble proteins compensate for the lower charges (z) of membrane proteins compared to soluble ones of similar MW. Obtaining the theoretical CCS for the structures of A β (1-42) tetramers derived from NMR restraints and that of two octamers models (β -barrel and β -sandwich) after gas phase simulations allowed us to establish that their values (MW, z, and CCS) were actually

in the same range as the ones of soluble proteins of similar MW. We then realized that the premise that membrane proteins present lower charges than their soluble counterparts of similar MW was not true for the structures under study, as the majority of chargeable residues are located at the N-termini not protected by the micelle at the time of ionization. Therefore, we decided to exclude larger calibrants (concanavalin A, alcohol dehydrogenase, pyruvate kinase, and glutamate dehydrogenase) leaving cytochrome C, β -lactoglobulin monomer/dimer and avidin as calibrants. Supplementary Fig. 20 shows that the experimental TWCCSN2 for A β (1-42) tetramers (orange) and A β (1-42) octamers (dark blue) are well interpolated within the regression line formed by the aforementioned smaller calibrants. This new calibration resulted in an increase in the experimental TWCCSN2 from 1160 Å² to 1598 Å² (+6) for A β (1-42) tetramers and from 1891 Å² to 2469 Å² (+8) for A β (1-42) octamers.

This increase matched the theoretical CCS from the gas phase simulated structure of A β (1-42) tetramers derived from NMR restraints (1657 Å²) and that of the β sandwich A β (1-42) octamer model (2546 Å²) (Fig. 4).

Reviewer 2

The manuscript by Ciudad et al. describes the structural characterization of Abeta(1-42) tetrameric and octameric structures. The structure of the tetramer is derived from NMR measurements. Cross-linking and SEC/IM-MS corroborate the findings of an Abeta tetramer, octamer, and other species. The study brings a 3D structure of an Abeta tetramer to the fore.

Point 1. The introduction is composed of only two paragraphs and lacking

background on A β or other oligomers that have been either modelled or determined experimentally. In short, the introduction would benefit from more background and context.

We have now extended the introduction providing more background on A β and other oligomers modelled and prepared experimentally (pages 4-5).

Point 2. A β oligomers were studied in dodecylphosphocholine (DPC), a detergent, and at non-physiological pH (8.5 and 9.5). The authors note in discussion that DPC is a membrane mimic and not a physiological environment. However, there is no mention in regards to pH. Would the proposed structures be stable at physiological pH?

Please see our answer to point 1 of reviewer 1.

Point 3. The 3D structure reveals an antiparallel arrangement with two of the three subunits exhibiting long flexible, solvent exposed N-termini. Given the structure, it would appear there would be large entropic barrier to insertion into the bilayer. Can the authors speculate on a plausible mechanism for insertion into the bilayer? Another issue not addressed, is the fact that charged and polar residues (such as His, QIn, and Lys) in their model (see Fig 1 and S17) are centrally located in the hydrophobic core of the bilayer. This simply does not make chemical sense and raises concerns to the validity of the proposed model. An explanation to rationalize exposure of such residues in the hydrophobic environment is warranted.

We have now rewritten the discussion and included a paragraph describing the potential mechanisms for insertion of A β (1-42) tetramers and octamers from a micelle to a lipid bilayer (pages 19-20) and another one detailing features that contribute to the stability of A β (1-42) tetramers and octamers in the lipid bilayer (page 20).

Point 4. In regards to the SEC/IM-MS measurements, increased activation conditions were used to disrupt the Ab complexes. The authors note tetramers broke into trimers and monomers, but this is not evident from the data presented (Fig S16 and S19). The assignment of oligomeric states by m/z and IM measurements is solid. The authors may consider isolating particular A β species using the quadrupole prior to subjecting them to increased activation? This has been done before (see doi: 10.1038/nature20820 and doi: 10.1007/s13361-016-1555-1), which would provide a more robust evaluation of complex gas-phase stability/dissociation.

We agree with the reviewer that as we are not selecting a single charge state with the quadrupole, SEC/IM-MS measurements at increased activation conditions do not provide enough evidence to conclude that tetramers break into trimers and monomers. However, our experience indicates that quadrupole selection is limited when coupling size exclusion chromatography directly to the mass spectrometer. Moreover, since the different charge states are contributed by more than one oligomer species, it would still remain challenging to prove this point. On this basis, we have decided to reformulate the text. Now, we do not report gas phase dissociation conclusions on A β (1-42) tetramers and only use these measurements to emphasize on the stability of A β (1-42) octamers: "These experiments allowed us to establish that octamers were not derived from the forced co-habitation of two tetramers in a micelle but rather from specific interactions between the A β subunits composing it." (page 11).

Point 5. IM measurements (CCS values) were compared to a limited set of A β models. The CCS values indicate a compact CCS value that agrees with models where the flexible loops have been removed from the structure prior to calculation. This approach raises significant concerns; deleting residues to match measured CCS values is not acceptable nor rigorous. First, flexible

regions can lead to an increase in CCS values (see DOI:10.1002/anie.201203047), which is not observed here. Does this imply that the N-termini are more ordered than flexible? Second, have other models been compared to the measured CCS values? Such as, a six-stranded cylindrin-like structure? Or other models? At present, additional models should be explored and also considering gas-phase minimization prior to CCS calculation.

As we have described in the answer of point 6 of reviewer 1, we have now carried out gas phase simulations prior to CCS calculations. We refer this reviewer to this answer.

We have carefully considered the use of additional models to be compared to the experimental CCS values of A β (1-42) tetramers and octamers but concluded that the A β (1-42) tetramer structure derived from NMR restraints and the β -barrel and β sandwich models for A β (1-42) octamers are the most relevant. Following is the list of considerations:

- 1. For the six-stranded A β (1-42) tetramer structure, we have only considered a flat and not a six-stranded cylindrin-like structure because we did not detect any NOEs for the amide protons of β 1 residues pointing outward of the β sheet core (page 7). For a cylindrin-like structure, we would expect detection of NOEs for those residues to indicate closure of the cylinder.*
- 2. SDS-PAGE analysis of non-boiled samples enriched in A β (1-42) octamers (β PFOsHIGH_A β (1-42)) led to a major band at 18 kDa as obtained for samples enriched in A β (1-42) tetramers (β PFOsLOW_A β (1-42)) (Fig. 3b and 3e). This result indicated that A β (1-42) octamers are derived from the assembly of two A β (1-42) tetramers. Consequently, we considered models for A β (1-42) octamers built from the assembly of two A β (1-42) tetramers.*
- 3. The use of different oligomer models previously reported in the literature such as a six-stranded cylindrin-like structure would require the use of different topologies from those established for the A β (1-42) tetramers in this work, and considered as the building block for A β (1-42) octamers. For example, the cylindrin-like model of A β is built from identical A β segments comprising residues 28-42 [Laganoswi, A. et al, Science 335 (2012)]. Instead, the*

structure of the A β (1-42) tetramers derived from this work is based on two different subunits comprising A β segments 9-42 and 29-42, respectively.

Point 6. No controls for the electrical recordings using planar lipid bilayers are presented. DPC is a detergent, which could disrupt the integrity of the bilayer giving rise to background signals. Were measurements of DPC in the absence of A β performed?

In our electrophysiology measurements, we performed controls using DPC in the absence of A β . In the previous version of the paper, we had the sentence “Controls were carried out to establish that the concentration of the detergent micelles present in the samples did not affect the stability of the bilayer” in the online Methods section entitled Electrical recordings with planar lipid bilayers. To make this important point clearer, we have now repeated the same sentence in the legend of Supplementary Fig. 22.

Point 7. The section titled “Preparation of ... enriched in A β (1-42) octamers” seems out of place. Suggest moving up and before the SEC-MS analysis section.

We have considered this remark carefully but concluded that it is important to first demonstrate the presence of A β (1-42) octamers in β PFOsA β (1-42) sample to rationalize establishing a protocol to prepare this species. Nevertheless, to clarify this point, we have added the sentence “Having detected A β (1-42) octamers in the β PFOsA β (1-42) sample, we attempted to enrich our sample in this oligomer form to pursue its characterization” at the beginning of the section “Preparation of a β PFOsA β (1-42) sample enriched in A β (1-42) octamers” on page 11.

Point 8. The manuscript could be improved by merging some figures. For example, Fig S17 can be incorporated into Fig S7.

We have now merged the two figures as Supplementary Fig. 8.

Point 9. For denatured and native MS measurements, the measured and theoretical masses should be reported along with error.

Supplementary Tables 3 and 4 now reports measured and theoretical masses for native and denatured MS measurements, respectively.

Point 10. Supplementary Fig. 18 is mislabelled. I presume the data shown is for PFOsHIGH not PFOsLOW.

The reviewer is correct. The data shown in Supplementary Fig. 18 is for β PFOsHIGH_A β (1-42). We have corrected this mistake.

Point 11. Pg 8, "... using DPC at natural abundance." This is unclear as DPC is non-natural.

In most of the NMR studies, to eliminate interfering proton signals in NMR spectra that come from the detergent itself, we used fully deuterated DPC, DPC-d38. However, to characterize the interaction of DPC molecules with the surface of A β (1-42) tetramer, we needed to use fully protonated DPC so we referred to it as DPC at natural abundance. To address the point made by the reviewer, we have now replaced "DPC at natural abundance" by "DPC at natural isotopic abundance" (page 9).

Point 12. Pg 15, "... we have precisely defined the structural .." Remove precisely.

We have now removed precisely from the sentence suggested by the reviewer.

Point 13. Although I am not able to expertly comment on the quality of the NMR data, the study seems well designed and executed.

We appreciate this comment from the reviewer.

Reviewer 3

This is a very interesting paper that presents a startling structure for the micelle-inserted amyloid-beta-42 polypeptide (AB42). AB42 is, of course, generally thought to be associated with the etiology of Alzheimer's disease. There is a body of evidence in the literature that AB42 can adopt some sort of transmembrane oligomer, with additional evidence that it can act (aberrantly) as an ion channel, potentially a mechanism that contributes to how AB42 promotes Alzheimer's disease. In this paper the authors used NMR spectroscopy to document that AB42 in dodecylphosphocholine micelles can form two closely related oligomeric structures. In one structure 4 subunits come together to form a single sheet with two subunits forming trans-micelle beta hairpins flank the two other subunits central in the tetrameric sheet as a single beta strands. What is amazing is that the 6-stranded tetrameric sheet does not form a closed barrel, but instead is an "open" sheet with solvent exposed edges for strands 1 and 6.

The other structure is a dimer of tetramers where the 6-stranded sheets form a sandwich in the micelle interior, again with exposed edges. The authors also present electrophysiology measurements in which ion channel activity is documented, apparently for both monomer and dimers, with the channel activity apparently being associated with the exposed edges of the monomeric or dimeric sheets.

Point 1. The NMR studies of this work were meticulously carried out and documented-- very convincing. Moreover, one can imagine how these structures might be stable in micelles: the horizontal diameter of the sheet is on the same order as the diameter of the micelle it sits in, so the need to form transmicelle beta barrels is relieved by the fact that the exposed edges of strands 1 and 6 are

not exposed to a bilayer interior, but instead water outside of the micelle water at the micelle-water interface. So, these structures seem reasonable as micelle structures.

However, it seems less clear that they could be energetically accommodated as transmembrane structures in a sealed lipid bilayer, where the exposed edges of strands 1 and 6 would be exposed only to lipid, resulting in burial backbone amide groups with unsatisfied H-bonding potential.

We have now rewritten the discussion and added a paragraph describing features that contribute to the stability of A β (1-42) tetramers and octamers in the lipid bilayer (page 20).

Point 2. The text, figures, and supporting material are all very well written. I think this work is EXTREMELY interesting and should be published following revision, without any requirement for additional experiments. However, the authors should provide more detail on what evidence they have that the micellar structure they carefully document in this work is maintained in lipid vesicles. This closely relates to the question of whether the channel activity they document really stems from AB42 channels that resemble the micellar structures. Again, more textual justification seems needed.

We have now rewritten the discussion and included a paragraph describing the potential mechanisms for insertion of A β (1-42) tetramers and octamers from a micelle to a lipid bilayer (pages 19-20).

Point 3. Finally, the sample composition, pH, and temperature need to be spelled out in detail in the caption to Figure 1.

As requested, we have now detailed the sample composition in the legend of Figure 1.

Reviewer 4

The manuscript by Ciudad et al deals with an important open question in our understanding of the Alzheimer's Disease (AD) Pathology cascade: the mechanism by which amyloid-beta causes damages neurons. The authors cite the "amyloid hypothesis" in which Abeta is assumed to be the primary toxic agent in AD, and note that the mechanism of toxicity is unclear. They also cite works in which soluble forms of Abeta can induce pores of various sorts in membranes. This is the background for their basic hypothesis: that Abeta structures can form pores in planar lipid bilayers, and therefore can serve as a mechanism of neurotoxicity. The primary focus of the work is in establishing the feasibility of this idea, and the authors perform detailed work in establishing this to be the case. The evidence that Abeta structures can form pores in planar lipid bilayers, and that this is specific to certain forms of Abeta, seems to be well established by this study, and supported by beautiful and clear figures.

I will address a different aspect of this study, which is the validity of their hypothesis: that Abeta can cause pores in the membranes of neurons adjacent to sufficient concentrations of Abeta, which will then disrupt neuronal activity. The principal evidence for that disruption is shown in Supplementary Figure 20, in which the membrane conductance is increased by the introduction of β PFOsA β (1-42) into planar lipid bilayers.

This makes a clear prediction of the effect of soluble Abeta on neuronal activity. The prediction is that the input conductance of these neurons will be increased. Intracellular recordings from several lines of APP+ transgenic mice (two of which are published), in which both soluble Abeta and amyloid plaques are present in the surrounding neuropil, by measurement of average input resistance in the neurons, do not show the increase in RIN that would be expected to be associated with such a conductance increase (Stern et al, J.

Neurosci 24, 4335-4340 (2004); Kellner et al, Neurobiology of Aging 35, 1982-1991 (2014)). I do not know of any studies which have shown different results. (In addition, measurement by several groups show hyperactivity in numerous lines of APP transgenic mice, which is the opposite effect that would be expected from a general conductance increase.) these are issues that should be addressed by the authors, if they wish to argue that increased membrane conductance is indeed a mechanism of neuronal disruption by Abeta.

NOTE: This, however, is not necessarily evidence that the conductance increase does not occur. The direct measurements of input resistance are made using electrodes inserted into the soma. It may well be that the toxic effects of Abeta on the membrane are local. Indeed, one study shows that there are specific effects on dendritic spines in proximity to plaques, where the concentrations of soluble Abeta are higher (Spires et al, J Neurosci. 25, 7278-7287 (2005)).

This is an hypothesis which can be tested, using targeted patch recordings in slices of APP transgenic mouse cortex. The recordings would have to target dendrites in mice in which plaque aggregation is severe, so that dendrites would have a high probability of proximity to plaques. While difficult, targeted recordings are performed on a daily basis by numerous laboratories. If the input conductance is increased in these dendrites, compared to aged-match recordings in WT mice, it would indeed be strong evidence that the hypothesis proposed by the authors is correct. In many studies, including the one I cited, there is evidence for stripping of spines from these dendrites, which I would predict will show the opposite effect: raising the local dendritic input resistance.

At present, however, there is no such evidence of which I am aware that the mechanism proposed by the authors disrupts neuronal activity in AD. And some evidence does exist that the overall conductance in the neuron is not raised in

APP-transgenic mouse model neurons. The current study does show an interesting possible mechanism of action of one of the forms of Abeta: one interesting enough to be further investigated. Again, evidence for the specific mechanisms (note the plural) of the pathophysiological effects of Abeta is a “missing link” in our understanding of the cascade of effects of the Abeta in the disease process.

Edward A. Stern, Brain Research Center, Bar Ilan University

We really appreciate the input given by find Prof. Stern to obtain in vivo evidence for the proposed mechanism of action of the A β (1-42) tetramers and octamers described in this work. Unfortunately, we have not been able to perform the proposed experiments as we find them beyond the scope of this paper. Nevertheless, we will consider them in future research.

In conclusion, we are grateful to the Reviewers for prompting these revisions, which we feel have contributed significantly to improving the paper. We very much hope that you now find the paper acceptable for publication.

Yours sincerely,

Natàlia Carulla

REVIEWERS' COMMENTS second round:

Reviewer #1 (Remarks to the Author):

The authors addressed the comments of all reviewers with great care. With regard to my comments I can confirm that they were also addressed satisfactorily. I have no further suggestions and recommend publication of the manuscript as is.

Reviewer #2 (Remarks to the Author):

The revised manuscript by Ciudad et al. has greatly improved. They have addressed my concerns and recommend publication.

Reviewer #3 (Remarks to the Author):

The authors have done a good job of addressing the reviewer concerns. As indicated in my original review I think this is an interesting, important, and novel work. I am in favor of publishing the revised manuscript as is.